# Fine-Grained Class-Conditional Distribution Balancing for Debiased Learning

**Miaoyun Zhao** *
Key Laboratory of Social Computing and Cognitive Intelligence
Dalian University of Technology
Liaoning, China

**Qiang Zhang** †
Key Laboratory of Social Computing and Cognitive Intelligence
Dalian University of Technology
Liaoning, China

## Abstract

Achieving group-robust generalization in the presence of spurious correlations remains a significant challenge, particularly when bias annotations are unavailable. Recent studies on Class-Conditional Distribution Balancing (CCDB) reveal that spurious correlations often stem from mismatches between the class-conditional and marginal distributions of bias attributes. They achieve promising results by addressing this issue through simple distribution matching in a bias-agnostic manner. However, CCDB approximates each distribution using a single Gaussian, which is overly simplistic and rarely holds in real-world applications. To address this limitation, we propose a novel Multi-stage data-Selective reTraining strategy (MST), which describes each distribution in greater detail using the hard confusion matrix. Building on these finer descriptions, we propose a fine-grained variant of CCDB, termed FG-CCDB, which enhances distribution matching through more precise confusion-cell-wise reweighting. FG-CCDB learns sample weights from a global perspective, effectively mitigating spurious correlations without incurring substantial storage or computational overhead. Extensive experiments demonstrate that MST serves as a reliable proxy for ground-truth bias annotations and can be seamlessly integrated with bias-supervised methods. Moreover, when combined with FG-CCDB, our method performs on par with bias-supervised approaches on binary classification tasks and significantly outperforms them in highly biased multi-class and multi-shortcut scenarios.

## 1 Introduction

Neural networks trained with standard Empirical risk minimization (ERM) Vapnik (1998) often suffer from spurious correlations: shortcuts that are predictive of the target class in the training data but irrelevant to the true underlying classification function LaBonte et al. (2023b). Samples exhibiting such spurious correlations typically dominate the training distribution and form the majority groups, while samples with different or conflicting correlations constitute the minority groups Radford et al. (2021). This imbalance across groups is also referred to as biased data, which results in poor ERM performance on the minority ones, sometimes even no better than random guessing Shah et al. (2020). Spurious correlations are prevalent in many high-stakes applications, including toxic comments identification Borkan et al. (2019), medical diagnosis Castro et al. (2020), and autonomous driving Pourkeshavarz et al. (2024), where both robustness and fairness are critical but overlooked by conventional methods. Take the traffic sign classification task as a vivid example Liu et al. (2023), in which the training data exhibits a strong bias: 99% of stop signs appear in red, whereas stop signs of other colors are rare and constitute a minority group Beery et al. (2018).

---

*miaoyun9zhao@gmail.com. Code: https://github.com/MiaoyunZhao/FG_CCDB.
†Corresponding author, zhangq@dlut.edu.cn.

Consequently, the classifier relies on the red color as a shortcut for recognizing stop signs, ignoring the textual "stop" features. This leads to biased predictions and poor generalization when the color cue is absent or misleading. Arjovsky et al. (2019); Geirhos et al. (2020); Beery et al. (2018). These challenges underscore the urgent need to develop classification methods that remain reliable across diverse data subgroups, especially in the presence of spurious correlations.

One of the most effective strategies for improving robustness against spurious correlations is to retrain models using group-balanced subsets derived from bias annotations Kirichenko et al. (2023). However, given the massive scale of modern datasets, manually labeling bias attributes is often prohibitively expensive, which motivates the development of annotation-free alternatives. Recent studies have shown that models trained with naïve ERM tend to favor biased solutions, which generalize poorly to minority groups — offering a "free lunch" for bias modeling Pezeshki et al. (2024); Puli et al. (2023). Accordingly, various methods have been developed to identify misclassified samples as belonging to minority groups. These approaches either explicitly highlight such samples or implicitly simulate group-balancing during the debiasing process to enhance group robustness LaBonte et al. (2023a); Pezeshki et al. (2024); Li et al. (2023a); Liu et al. (2021). However, they often rely on empirically chosen hyperparameters to control the upweighting of minority groups, which can easily lead to overemphasis on these groups and, in turn, degrade performance on the majority ones. As a result, held-out annotations are often required for effective hyperparameter tuning. Recent research on Class-conditional distribution balancing (CCDB) Zhao et al. (2025) reveals that spurious correlations arise from the mismatches between class-conditional and marginal distributions (usually caused by bias cues), and addresses it by reweighting samples to minimize the mutual information between bias cues and class labels without hyperparameter searching. However, CCDB performs coarse distribution matching by treating each distribution as a single Gaussian, which rarely holds in real-world applications. In practice, instances within the same class often exhibit multi-modal distributions due to hidden bias cues. Thus, this coarse matching fails to capture intra-class variations, leaving residual spurious correlations unaddressed.

To resolve these limitations, we propose a fine-grained distribution matching technique based on CCDB, termed Fine-Grained Class-Conditional Distribution Balancing (FG-CCDB), which achieves stronger mitigation of spurious correlations without relying on bias annotations. Our approach is developed from two key perspectives: ($i$) **Fine-grained distribution description.** Inspired by the "free lunch" phenomenon in ERM — where models tend to overfit to spurious correlations — we introduce a Multi-stage data-Selective reTraining strategy (MST) for bias characterization, which capable of tackling multi-shortcuts by relate the hard confusion matrix to bias-aligning and conflicting partitions, and employing a multi-stage, data-selective retraining strategy to enhance the reliability of these partition assignments, which iteratively refines predictions from the overfitted model. This process yields a confusion matrix that approximates the ground-truth group partition when spurious correlations arise from a single shortcut. ($ii$) **Fine-grained distribution matching.** Building on the confusion matrix identified by MST, we extend CCDB into a fine-grained formulation, termed Fine-Grained Class-Conditional Distribution Balancing (FG-CCDB). It provides a discrete multi-modal approximation of both class-conditional and marginal distributions, enabling precise mode-wise alignment and thus more thorough mitigation of spurious correlations than the original CCDB. **The main contributions of this work are as follows:** ($i$) We propose an annotation-free bias exploration method with multi-stage refinement, based on model overfitting, which generalizes beyond singular shortcut and serves as a reliable alternative to human annotations. ($ii$) We introduce FG-CCDB, a lightweight and scalable debiasing method that enables fine-grained mode-wise reweighting and is well-suited for multi-class classification and multi-shortcut mitigation. ($iii$) Extensive experiments show that our method matches or surpasses bias-supervised baselines, achieving strong performance without requiring bias annotations.

## 2 RELATED WORK

The related work is structured around the two core aspects of our contribution.

### 2.1 BIAS EXPLORATION

Primary approaches define bias as texture Bahng et al. (2020), background Venkataramani et al. (2024), or image style Li et al. (2025)—features presumed irrelevant to class labels. These methods

often rely on tailored architectures or training schemes to detect specific bias cues Hong & Yang (2021), but generalize poorly to unknown biases. To overcome this, recent data-driven strategies interpret bias as group imbalance or latent substructures. Some methods, like JTT Liu et al. (2021), LfF Nam et al. (2020), and RIDGE Pezeshki et al. (2024), identify bias via consistently misclassified (hard) samples under ERM. Others rely on model disagreement, *e.g.,* DebiAN Li et al. (2022b) iteratively trains a bias "discoverer" alongside a main classifier, XRM Pezeshki et al. (2024) uses a pair of biased auxiliary models to generate pseudo group labels across the training set, DDB Ciranni et al. (2025) utilizes a diffusion model to generate bias-aligned data, which amplifies the bias reliance. Other methods, such as GEORGE Sohoni et al. (2020), apply unsupervised feature clustering to decompose each class into latent subgroups. Few of these methods conduct a thorough evaluation on the quality of bias prediction. Another trend leverages vision-language models (e.g., CLIP Radford et al. (2021)) to infer explainable bias attributes Jain et al. (2023); Kim et al. (2024); Wiles et al. (2022), though they are constrained by predefined vocabularies and may miss unexpected biases.

## 2.2 BIAS MITIGATION

**Bias annotation dependent**. With the assistance of bias annotations, a variety of methods have been developed to mitigate spurious correlations. GroupDRO Sagawa et al. (2020) groups data based on class and bias annotations and optimizes for the worst-group performance. DFR Kirichenko et al. (2023) improves robustness by retraining only the last layer using a small, balanced validation set. MAPLE Zhou et al. (2022) uses a measure based on validation set with explicit bias annotations to reweight training samples. LISA Yao et al. (2022) utilizes data augmentation technique to encourage bias-invariant features. Though effective, relying on costly bias annotations limits their scalability in real applications.

**Bias-conflicting samples dependent.** To mitigate spurious correlations without manual annotation, recent studies often leverage disagreements among auxiliary models to identify bias-conflicting samples and focus learning on them.Nam et al. (2020); Liu et al. (2023); Chu et al. (2021); Liu et al. (2021). To better identify bias-conflicting samples, SELF LaBonte et al. (2023b) proposes to split the training data and applying early stopping for effective bias-conflicting detection. uLA Tsirigotis et al. (2023b) leverages pretrained self-supervised models to extract bias-relevant information. DPR Han et al. (2024) uses the Generalized cross-entropy loss Nam et al. (2020) to amplify model bias. However, these methods rely on empirically tuned parameters—often requiring a split of annotated subsets—and their simple binary partitioning into bias-aligned and bias-conflicting samples is insufficient to fully capture the structure of bias, ultimately limiting generalization.

**Bias-agnostic.** Beyond bias-aware techniques, several bias-agnostic approaches have emerged, motivated by diverse perspectivesPuli et al. (2023); Jain et al. (2024). MASKTUNE Asgari et al. (2022), ExMap Chakraborty et al. (2024), and DaC Noohdani et al. (2024) reduce reliance on spurious features by identifying bias-related regions via heatmaps, which restricts their applicability to the image domain. Stable learning approachesZhang et al. (2021); Yu et al. (2023) treat spurious correlations as effects of unknown confounders and attempt to mitigate them by decorrelating features, though this is difficult to achieve in practice. GERNE Asaad et al. (2025) leverages the gradient differences between two batches to identify a debiasing direction, along which the model is optimized. CCDB Zhao et al. (2025) seeks to mitigate spurious correlations by minimizing the mutual information between spurious features and class labels via distribution matching. Although effective, its coarse matching strategy limits generalization performance.

## 3 OUR METHOD

Our work builds on the existing method CCDB, which attributes spurious correlations to distribution mismatches and addresses them through sample reweighting without requiring bias annotations. However, CCDB performs distribution matching in a relatively coarse manner by modeling each distribution as a single Gaussian. To enable more accurate alignment–and thereby more effective spurious correlations elimination–we propose a fine-grained extension. Specifically, we introduce a multi-stage data-selective retraining strategy (MST) that characterizes bias structure via the hard confusion matrix, allowing for a discrete multi-modal description of each distribution. Based on these multi-modal distributions, we develop Fine-Grained Class-Conditional Distribution Balancing (FG-CCDB), which performs alignment at the mode level.

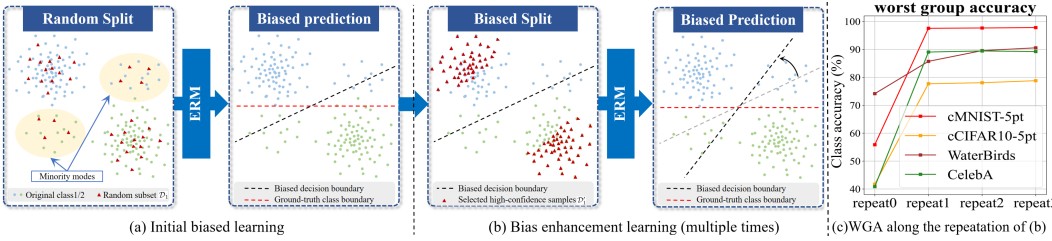

Figure 1: The MST framework progresses from an initial partition with limited minority group coverage (a) to a more complete identification in stage (b). (c) WGA under different bias-capturing qualities.

We consider the task of predicting a label $y \in \mathcal{Y}, \mathcal{Y} = \{1, \ldots, C\}$ based on an input $\boldsymbol{x} \in \mathcal{X}, \mathcal{X} \subset \mathbb{R}^d$. Following prior work, we define a *shortcut* as an explainable attribute (*e.g.,* color, background) that is spuriously correlated with class labels and highly predictive, and focus on a more general setting in which each data point $(\boldsymbol{x}, y)$ may be associated with one or more shortcuts. Motivated by Tsirigotis et al. (2023a), we use an auxiliary biased model to predict the bias label $s \in \mathcal{S}$, which share the same label space as $y$, *i.e.,* $|\mathcal{S}| = |\mathcal{Y}|$. Note that our goal is for $s$ to capture general and harmful bias information that humans may not preconceive Li et al. (2022b), rather than only physically interpretable attributes. The value $s$ represents spurious signals that an ERM model prefers over core features and that consequently cause evaluation failures. $s = i$ denotes all spurious cues that cause samples from other classes to be misclassified as class $i$. These cues may correspond to interpretable shortcuts, combinations of multiple shortcuts, or entangled, uninterpretable patterns. By combining $s$ and $y$, we partition the dataset into modes $\mathcal{M} = \mathcal{S} \times \mathcal{Y}$, which exactly corresponde to the hard confusion matrix. When the bias corresponds to a single shortcut, this reduces to conventional group partitions. To distinguish our data partitioning approach from traditional group-based methods, we refer to the partitions derived from the confusion matrix as **modes**. Accordingly, diagonal entries represent majority (bias-aligning) modes, and off-diagonal entries correspond to minority (bias-conflicting) modes. With the confusion matrix, one can infer a discrete multi-modal approximation of both the class-conditional and marginal distributions over bias information. The goal is twofold: $(i)$ to train a biased model that can effectively explore the underlying bias cues; $(ii)$ to train a debiased model that invariant to bias information and achieves uniform performance across all modes.

## 3.1 BIAS EXPLORATION THROUGH OVERFITTING

In this section, we introduce the proposed multi-stage data-selective retraining (MST) technique and demonstrate its compatibility with existing bias-supervised methods. It is well established that, in the presence of spurious correlations, ERM tends to overfit to majority groups in training data, leading to an over-reliance on bias cues and poor generalization to minority groups. Recent studies LaBonte et al. (2023b); Tsirigotis et al. (2023b) have made preliminary attempts to exploit this overfitting behavior to mitigate spurious correlations, revealing that the predictions of overfitted models are strongly aligned with bias cues. Furthermore, Lee et al. (2023); Ciranni et al. (2025) find that removing bias-conflicting samples improves bias prediction and point out that, in principle, if all bias-conflicting samples were removed, one could train a bias-capturing model that provides ideal learning signals for debiasing. Inspired by these insights, we propose a multi-stage framework for refined bias prediction, which further leverages model overfitting and serves as an approximate substitute for human annotations. The overall framework consists of two basic stages (Figure 1(a)(b)): initial bias learning and bias enhancement learning. The first stage extracts primary bias patterns, while the second amplifies them in the model's predictions, yielding a reliable bias predictor.

**Initial bias learning.** Given an accessible train dataset $\mathcal{D} := \{(\boldsymbol{x}_i, y_i)\}_{i=1}^N$ of $N$ samples and $C$ classes. Following prior works Zhao et al. (2025); LaBonte et al. (2023b); Pezeshki et al. (2024), we explore bias information by randomly splitting $\mathcal{D}$ into two subsets $\mathcal{D}_1$ and $\mathcal{D}_2$, where $\mathcal{D}_1$ contains a fraction $\gamma$ of the original data (Figure 1(a), left). We then perform naïve ERM on $\mathcal{D}_1$ to train a biased model $f_{\boldsymbol{\theta}_1}$, which typically performs well on majority groups but poorly on minority groups. Unlike prior worksLaBonte et al. (2023b); Pezeshki et al. (2024), which use $95\%/50\%$ of the data for biased training, our goal is to maximize the model's alignment with bias cues to better reveal underlying spurious correlations. As demonstrated in our experiments (Figure 3(right)), a smaller $\gamma$ proves more effective for bias exploration, with $\gamma = 10\%$ emerging as a sweet spot. Since the data is

Figure 2: (a) The framework of our FG-CCDB (b) The joint distribution $\mathbf{J}$ of bias and class labels estimated by our method. (c) Toy example to show how FG-CCDB differs from group balancing.

randomly split, some samples from minority modes inevitably participate in training, which weakens the model's tendency to align its predictions with bias cues (Figure 1(a) right). To counteract this effect, we introduce a subsequent amplification stage.

**Bias enhancement learning.** Amplifying bias in model predictions is non-trivial. Our key idea is to guide the bias prediction model to focus exclusively on majority modes, *i.e.,* to construct a training subset $\mathcal{D}_1$ that contains little to no samples from minority modes. This idea of removing bias-conflicting samples has been shown to effectively amplify bias in prior work Lee et al. (2023); Ciranni et al. (2025). Such a setup forces the model to overfit to the majority modes and align more strongly with the corresponding bias cues, thereby behaving like a bias predictor and exhibiting near-zero generalization ability on minority modes. To achieve this, we introduce a data selection procedure based on the predictions of $f_{\boldsymbol{\theta}_1}$, forming an extremely biased subset $\mathcal{D}_1'$, on which a more biased model is trained. Specifically, for each sample $(\boldsymbol{x}_i, y_i)$ in $\mathcal{D}$, we infer the softmax output as $\boldsymbol{h}_i = f_{\boldsymbol{\theta}_1}(\boldsymbol{x}_i)$. Within each class, we select the top $\beta$ fraction of samples ($\beta \in [0, 1]$) with the highest prediction confidence (measured by $\boldsymbol{h}_i$), and aggregate them to form $\mathcal{D}_1'$. In our experiments, we find that $\beta = 50\%$ offers a stable and reliable choice. Since $f_{\boldsymbol{\theta}_1}$ is biased toward majority modes, the high-confidence samples are more likely to come from those modes. Consequently, $\mathcal{D}_1'$ filters out most minority mode instances and is thus more biased than $\mathcal{D}_1$. (Figure 1(b) left). We then train a new biased model $f_{\boldsymbol{\theta}_2}$ using naïve ERM on $\mathcal{D}_1'$. The resulting model serves as the final bias predictor to produce bias labels for $\mathcal{D}$. Combined with the target class labels, the resulting hard confusion matrix yields estimated mode partitions over the space $|\mathcal{S}| \times |\mathcal{Y}|$, which can serve as a proxy for group annotations in bias-supervised methods (Figure 1(b) right).

Notably, the "Bias enhancement learning" stage can be repeated to further improve bias prediction accuracy. Only the biased model from the final repetition is used to generate bias labels. As shown in the experiments(Figure1(c)), a single iteration already achieves performance comparable to existing methods, while further iterations lead to gradually converging performance with diminishing gains.

### 3.2 FINE-GRAINED CLASS-CONDITIONAL DISTRIBUTION BALANCING

In this section, we present the Fine-grained Class-Conditional Distribution Balancing (FG-CCDB) approach. With the hard confusion matrix obtained via MST, FG-CCDB improves both the quality of distribution matching and the efficiency of sample reweighting.

The original CCDB proposes to mitigate spurious correlations by directly minimizing the mutual information between bias cues and target classes, which is achieved by aligning each class-conditional distribution with the marginal distribution, while simultaneously balancing class proportions — a generalization to traditional class balancing technique. Specifically, the objective is to minimize:

$$\mathcal{L}_{\boldsymbol{\omega}} = I(\boxed{z}, y) - H(y) = \mathbb{E}_{p_{\boldsymbol{\omega}}(y)} D_{\mathrm{KL}}[p_{\boldsymbol{\omega}}(\boxed{z}|y) \| p(\boxed{z})] + \mathbb{E}_{p_{\boldsymbol{\omega}}(y)} \log p_{\boldsymbol{\omega}}(y) \tag{1}$$

where $\boxed{z}$ denotes the latent feature (with gradients detached) extracted by the biased model prior to the fully connected layer, which predominantly captures bias cues. $\boldsymbol{\omega}$ denotes the sample weights to be optimized, and $D_{\mathrm{KL}}[\cdot \| \cdot]$ refers to the Kullback–Leibler divergence Kullback & Leibler (1951). Since the true distributions associated with $\boldsymbol{z}$ and $y$ are unknown, CCDB approximates them using single Gaussian, which is insufficient for complex data with inherently multi-modal structures. Moreover, CCDB's sample-level reweighting requires storing and processing feature representations for the entire dataset, incurring additional computational cost.

Our work adopts the same objective as Equation 1. To overcome the aforementioned limitations, we derive a discrete multi-modal approximation of both class-conditional and marginal distributions from the hard confusion matrix, which enables localized, mode-wise distribution matching and leads

to more accurate and scalable reweighting. As shown in Figure 2 (a), we represent the confusion matrix as $\mathbf{M} \in \mathbb{R}^{C \times C}$, where $\mathbf{M}_{i,j}$ denotes the number of samples belonging to mode $(s, y) = (i, j)$, Thus, the joint distribution over $(\boldsymbol{z}, y)$ is approximated with a discretized version over modes $(s, y)$, which is characterized by matrix $\mathbf{J} \in \mathbb{R}^{C \times C}$ with $\mathbf{J}_{i,j} = \frac{\mathbf{M}_{i,j}}{N}$ represents the probability of a sample belonging to mode $(s, y) = (i, j)$, $N$ is the total number of training samples. By design, we define a class-conditional distribution matrix $\mathbf{P} \in \mathbb{R}^{C \times C}$ such that the $j$-th column $\mathbf{P}_{:,j}$ encodes $p(\boldsymbol{z}|y = j)$, and a marginal distribution vector $\boldsymbol{q} \in \mathbb{R}^C$ that captures $p(\boldsymbol{z})$. Both $\mathbf{P}$ and $\boldsymbol{q}$ are computed directly from $\mathbf{J}$ as follows:

$$p(\boldsymbol{z}|y = j) \stackrel{\text{def}}{\approx} \mathbf{P}_{:,j} = \frac{\mathbf{J}_{:,j}}{\sum_i \mathbf{J}_{i,j}}, \qquad p(\boldsymbol{z}) \stackrel{\text{def}}{\approx} \boldsymbol{q} = \sum_j \mathbf{J}_{:,j} \tag{2}$$

Figure 2 (b) shows the joint distribution matrix $\mathbf{J}$ estimated by our MST across four datasets. Clear spurious correlations are observed, as evidenced by the strong diagonal elements (aligned along the yellow line), which indicate a high dependency between labels and bias cues. To eliminate these spurious correlations and minimize equation1, we introduce mode-level weighting parameter $\mathbf{W} \in \mathbb{R}^{C \times C}$ to adjust each class-conditional distribution so that it aligns with the marginal distribution. A straightforward solution for $\mathbf{W}$ is,

$$\mathbf{W}_{i,j} = \frac{\boldsymbol{q}_i}{\mathbf{P}_{i,j}}, \qquad \text{for } i, j = 1, \cdots, C \tag{3}$$

Note, equation 3 achieves exact distribution matching, *i.e.,* $\mathbf{W}_{:,j} \odot \mathbf{P}_{:,j} = \boldsymbol{q}$, meaning that all class-conditional distributions are reweighted to align with the same marginal distribution $\boldsymbol{q}$, where $\odot$ denotes the Hadamard product. For a given mode $(s, y) = (i, j)$, assuming uniform contribution from its samples, the corresponding sample weight is,

$$\boldsymbol{w}_{i,j} = \frac{\mathbf{W}_{i,j}}{\mathbf{M}_{i,j}}, \qquad \text{for } i, j = 1, \cdots, C \tag{4}$$

Note that beyond distribution matching, Equation 4 inherently solved the class imbalance issue: the mode with more samples gets smaller weights. As a result, FG-CCDB simultaneously minimizes both terms in Equation 1. These weights are subsequently used during debiasing to reweight training samples according to their mode identities.

It is worth noting that *our distribution matching fundamentally differs from conventional group balancing* (see example in Figure2(c)): ($i$) Unlike group balancing, which aims to reduce differences across all entries in the mode matrix, our method focuses solely on aligning the class-conditional distributions—i.e., reducing the variation among columns in $\mathbf{P}$—while preserving intra-column imbalance. This allows for more flexible training by merely minimizing mutual information rather than enforcing strict equality. ($ii$) By minimizing the divergence between conditional and marginal distributions, our method and CCDB implicitly achieve "covariate balance" from the view of causal inference, specifically, by finding a reweighting that makes the confounder (bias) independent of the treatment (core feature), ultimately forcing the statistical model to rely solely on core features for inference Neal (2020). ($iii$) Simple scale balancing between majority and minority modes is insufficient for generalization, as majority modes typically exhibit greater diversity. Our method applies a more aggressive reweighting strategy. For example, the ratio between the largest and smallest mode weights in FG-CCDB reaches 1000, compared to just 100 in conventional group balancing. *Compared to CCDB, our sample reweighting approach offers several key advantages*: ($i$) It performs distribution matching across multiple localized regions defined by the confusion matrix, enabling more precise alignment and more thorough removal of spurious correlations; ($ii$) The sample weights are computed efficiently in closed form, without requiring any iterative optimization; ($iii$) Instead of assigning weights individually to each sample, FG-CCDB assigns a shared weight to samples within the same mode, resulting in negligible computational and memory overhead.

After completing MST and FG-CCDB, we train a debiased model $f_\phi$ by incorporating sample weights into the data sampling process using PyTorch's "*torch.utils.data.WeightedRandomSampler*" following Zhao et al. (2025). Unless otherwise specified, we refer to the entire procedure as FG-CCDB for brevity. A full algorithm of the proposed method is provided in Appendix A.

Table 1: Classification performance on real-world datasets. We report the average test accuracy(%) and std.dev. over 5 random seeds. Best bias-agnostic results in bold.

| Methods | Bias label | | Waterbirds | | CelebA | | CivilComments | |
|---|---|---|---|---|---|---|---|---|
| | Train | Val | i.i.d. | WGA | i.i.d. | WGA | i.i.d. | WGA |
| GroupDRO | Yes | Yes | 93.50 | 91.40 | 92.90 | 88.90 | 84.2 | 73.7 |
| DFR | Yes | Yes | 94.20±0.4 | 92.90±0.2 | 91.30±0.3 | 88.30±1.1 | 87.2±0.3 | 70.1±0.8 |
| LfF | No | Yes | 97.50 | 75.20 | 86.00 | 77.20 | 68.2 | 50.3 |
| JTT | No | Yes | 93.60 | 86.00 | 88.00 | 81.10 | 83.3 | 64.3 |
| LC | No | Yes | - | 90.50±1.1 | - | 88.10±0.8 | - | 70.30±1.2 |
| SELF | No | Yes | - | 93.00±0.3 | - | 83.90±0.9 | - | 79.10±2.1 |
| DaC | No | Yes | 95.3±0.4 | 92.3±0.4 | 91.4±1.1 | 81.9±0.7 | - | - |
| ERM | No | No | **97.30** | 72.60 | **95.60** | 47.20 | 81.6 | 66.7 |
| MASKTUNE | No | No | 93.00±0.7 | 86.40±1.9 | 91.30±0.1 | 78.00±1.2 | - | - |
| uLA | No | No | 91.50±0.7 | 86.10±1.5 | 93.90±0.2 | 86.50±3.7 | - | - |
| XRM | No | No | 90.60 | 86.10 | 91.0 | 88.5 | 83.5 | 70.1 |
| DebiAN | No | No | 90.80 | 78.19 | 84.0 | 52.9 | - | - |
| DDB | No | No | - | 90.34 | - | - | - | - |
| GERNE | No | No | - | 89.88±0.67 | - | 74.24±2.51 | - | 63.10±0.22 |
| CCDB | No | No | 92.59±0.10 | 90.48±0.28 | 90.08±0.19 | 85.27±0.28 | 83.60±0.21 | 75.00±0.26 |
| **FG-CCDB** | No | No | 92.50±0.52 | **90.56±0.24** | 89.71±0.54 | **89.22±0.19** | **86.99±0.14** | **78.52±0.42** |

Table 2: Results on UrbanCars.

| Methods | Bias label | | I.D. | Gap due to shortcuts(↑) | | |
|---|---|---|---|---|---|---|
| | Train | Val | Acc | BG | CoObj | BG+CoObj |
| GroupDRO | Yes | Yes | 91.6 | -10.9 | -3.6 | -16.4 |
| JTT | No | Yes | 95.9 | -8.1 | -13.3 | -40.1 |
| DaC | No | No | 98.17 | -3.78 | -9.78 | -58.58 |
| ERM | No | No | 97.6 | -15.3 | -11.2 | -69.2 |
| ExMap | No | No | - | -5.9 | -9.9 | -30.7 |
| DebiAN | No | No | 98.0 | -14.9 | -10.5 | -69.0 |
| DDB | No | No | 86.39 | **-1.85** | **-0.52** | **-0.12** |
| **FG-CCDB** | No | No | 92.98 | -4.17 | -7.37 | -4.9 |

Table 3: Ablation study on four datasets.

| Methods | Waterbirds | | CelebA | | cMNIST | cCIFAR10 |
|---|---|---|---|---|---|---|
| | i.i.d. | WGA | i.i.d. | WGA | i.i.d. | i.i.d. |
| GroupDRO | 93.50 | 91.40 | 92.90 | 88.90 | 84.20 | 57.32 |
| GroupDRO-MST | 90.82±0.05 | 88.47±0.35 | 88.69±0.15 | 85.21±0.02 | 84.07±0.22 | 55.73±0.54 |
| DFR | **94.20±0.4** | **92.90±0.2** | 91.30±0.3 | 88.30±1.1 | - | - |
| DFR-MST | 92.53±0.50 | 91.49±0.72 | 88.80±0.20 | 85.87±0.29 | - | - |
| FG-CCDB | 92.50±0.52 | 90.56±0.24 | 89.71±0.54 | **89.22±0.19** | 98.21±0.02 | 78.06±0.30 |
| FG-CCDB-sup | 91.54±0.11 | 91.76±0.13 | **93.14±0.16** | 89.09±0.12 | **98.26±0.21** | **78.53±0.37** |

## 4 EXPERIMENTAL RESULTS

In this section, we demonstrate the effectiveness of our method from five perspectives: ($i$) We conduct experiments on real-world binary classification benchmarks with either single or multiple shortcuts, such as Waterbirds Zhou et al. (2022), CelebA Zhou et al. (2022), CivilComments Koh et al. (2021), and UrbanCars Li et al. (2023b) to validate the overall effectiveness of our method; ($ii$) We further evaluate our method on challenging multi-class datasets, including cMNIST Li et al. (2022a) and cCIFAR10 Hendrycks & Dietterich (2018) to assess its robustness under highly biased conditions; ($iii$) To evaluate the reliability of the bias cues explored by MST, we compare them with ground-truth bias annotations and analyze the effects of repeating the "bias enhancement learning" procedure; ($iv$) we conduct an ablation study to demonstrate that each technical component (MST and FG-CCDB) makes a distinct and independent contribution to the final performance. ($v$) Finally, we analyze the effects of hyperparameters ($\gamma$ and $\beta$) on the performance of MST.

For all datasets, we adopt the same train-validation-test split following Liu et al. (2021); Tsirigotis et al. (2023b) for fair comparison. Results are averaged over 5 random seeds, and for each seed, the best-performing model (the one with the highest worst-class accuracy on the validation set) is selected Tsirigotis et al. (2023b). Unless otherwise stated, we repeat the "bias enhancement learning" process three times for FG-CCDB. See the appendix for the detailed experimental setup.

**Compared methods.** To demonstrate the superiority of our method in addressing spurious correlations and its potential to serve as an approximate substitute for bias-supervised methods, we compare it with both bias-supervised and bias-agnostic techniques. GroupDRO Sagawa et al. (2020) and DFR Kirichenko et al. (2023) are fully bias-supervised during both training and validation, and serve as strong baselines. LfF Nam et al. (2020), JTT Liu et al. (2021), LC Liu et al. (2023), DaC Noohdani et al. (2024), and SELF LaBonte et al. (2023b) rely on pseudo-bias supervision during training, but still require bias annotations during validation to achieve optimal performance. In contrast, ERM, uLA Tsirigotis et al. (2023b), MASKTUNE Asgari et al. (2022), XRM Pezeshki et al. (2024), DebiAN, ExMap, DDB Ciranni et al. (2025), GERNE Asaad et al. (2025), and CCDB Zhao et al. (2025), similar to our method, are entirely bias-agnostic throughout both training and validation.

### 4.1 BINARY CLASSIFICATION WITH A SINGLE OR MULTIPLE SHORTCUTS

The results on real-world binary classification with a single shortcut are shown in Table1. Although i.i.d. performance reflects overall accuracy, it can mask disparities across groups. In contrast, worst-

Table 4: Results on cMNIST and cCIFAR10 with various bias-conflicting ratios in the training set. The test accuracy(%) is averaged over 5 random seeds.The best results are indicated in bold.

| Methods | Bias label | | cMNIST | | | | cCIFAR10 | | | |
|---|---|---|---|---|---|---|---|---|---|---|
| | Train | Val | 0.5% | 1% | 2% | 5% | 0.5% | 1% | 2% | 5% |
| GroupDRO | Yes | Yes | 63.12 | 68.78 | 76.30 | 84.20 | 33.44 | 38.30 | 45.81 | 57.32 |
| LfF | No | Yes | $52.50_{\pm2.43}$ | $61.89_{\pm4.97}$ | $71.03_{\pm2.44}$ | $80.57_{\pm3.84}$ | $28.57_{\pm1.30}$ | $33.07_{\pm0.77}$ | $39.91_{\pm0.30}$ | $50.27_{\pm1.56}$ |
| LC | No | Yes | $71.25_{\pm3.17}$ | $82.25_{\pm2.11}$ | $86.21_{\pm1.02}$ | $91.16_{\pm0.97}$ | $34.56_{\pm0.69}$ | $37.34_{\pm0.69}$ | $47.81_{\pm2.00}$ | $54.55_{\pm1.26}$ |
| DaC | No | Yes | 53.24 | 75.02 | 87.60 | 94.70 | 21.01 | 28.01 | 36.56 | 51.06 |
| ERM | No | No | $35.19_{\pm3.49}$ | $52.09_{\pm2.88}$ | $65.86_{\pm3.59}$ | $82.17_{\pm0.74}$ | $23.08_{\pm1.25}$ | $25.82_{\pm0.33}$ | $30.06_{\pm0.71}$ | $39.42_{\pm0.64}$ |
| uLA | No | No | $75.13_{\pm0.78}$ | $81.80_{\pm1.41}$ | $84.79_{\pm1.10}$ | $92.79_{\pm0.85}$ | $34.39_{\pm1.14}$ | $62.49_{\pm0.74}$ | $63.88_{\pm1.07}$ | $74.49_{\pm0.58}$ |
| GERNE | No | No | $77.25_{\pm0.17}$ | $83.98_{\pm0.26}$ | $87.41_{\pm0.31}$ | $90.98_{\pm0.13}$ | $39.90_{\pm0.48}$ | $45.60_{\pm0.23}$ | $50.19_{\pm0.18}$ | $56.53_{\pm0.32}$ |
| CCDB | No | No | $83.20_{\pm2.17}$ | $87.95_{\pm1.59}$ | $91.02_{\pm0.28}$ | $96.37_{\pm0.25}$ | $55.07_{\pm0.85}$ | $63.28_{\pm0.46}$ | $67.78_{\pm0.78}$ | $74.64_{\pm0.34}$ |
| **FG-CCDB** | No | No | $\mathbf{89.02}_{\pm0.45}$ | $\mathbf{94.93}_{\pm0.17}$ | $\mathbf{96.18}_{\pm0.19}$ | $\mathbf{98.21}_{\pm0.02}$ | $\mathbf{55.28}_{\pm0.54}$ | $\mathbf{64.66}_{\pm0.48}$ | $\mathbf{71.69}_{\pm0.31}$ | $\mathbf{78.06}_{\pm0.30}$ |

group accuracy (WGA) directly measures robustness by focusing on the most challenging subpopulations. With bias annotations available during both training and validation, GroupDRO and DFR demonstrate strong generalization performance on the worst group, serving as a challenging upper bound. In contrast, methods that only use bias annotations during validation show a bit inferior performance. The situation becomes more challenging when access to bias annotations is not permitted. In this case, existing bias-agnostic methods consistently fall short of the supervised upper bound on at least one of the datasets. Remarkably, SELF, CCDB and our method surpass the supervised upper bound on CivilComments by a large margin. This is because they apply stronger upweighting to the minority groups/modes. Among all compared methods, including those with full supervision, our method consistently achieves the best or competitive WGA across all three datasets, highlighting its effectiveness in eliminating the need for human annotations.

Table2 presents the results on UrbanCars with multiple shortcuts: background (BG) and co-occurring object (CoObj). The in-distribution accuracy(I.D. Acc) and gap-related metrics are adopted from Li et al. (2023b)(See appendix for details). The BG/CoObj/BG+CoObj Gap is the drop in accuracy between mean and cases when only the BG/CoObj/BG+CoObj is uncommon. A smaller drop indicates better generalization. On average, BG+CoObj is the most challenging one and most compared methods suffer a significant drop on it. GroupDRO can mitigate multiple shortcuts; however, they require access to labels of both shortcuts. Although DDB shows the smallest overall drops across all bias-conflicting scenarios, its base I.D. Acc is the lowest among all compared methods. Overall, our method consistently achieves the best balance between high I.D. Acc and small drops compared to other bias-agnostic methods (particularly on the challenging BG+CoObj generalization). It performs comparably to, or better than, methods that rely on bias annotations. These results confirm that our approach provides a general framework for handling multi-shortcut scenarios. Please refer to Appendix D for more details.

## 4.2 MULTI-CLASS CLASSIFICATION UNDER EXTREME SPURIOUS CORRELATIONS

In this section, we use the synthetic datasets cMNIST and cCIFAR10 to evaluate the effectiveness of our method under challenging multi-class settings with extreme spurious correlations. For each, we vary the ratio of bias-conflicting samples in the training set to control the strength of spurious correlations and evaluate performance on a completely unbiased test set. Following Tsirigotis et al. (2023b), the bias-conflicting ratios are set to $\{0.5\%, 1\%, 2\%, 5\%\}$ for both datasets, where $0.5\%$ indicates an extremely biased scenario. The generalization accuracies are reported in Table 4. We observe that: ($i$) On both datasets, our method consistently achieves the best performance. In particular, it outperforms the second-best method by a large margin on cMNIST; ($ii$) On cCIFAR10, the improvements become more pronounced as the bias-conflicting ratio increases (i.e., at $2\%$ and $5\%$).

A comparison of the results in Table 1 and Table 4 reveals a phenomenon similar to that reported in Zhao et al. (2025): bias-supervised methods tend to perform well on basic binary classification tasks, whereas bias-agnostic methods are relatively more effective in complex multi-class classification scenarios. In contrast, CCDB demonstrates strong performance across both scenarios. With fine-grained distribution matching, FG-CCDB further boosts performance over CCDB by a significant margin, highlighting the effectiveness of more thorough spurious correlations elimination.

To demonstrate the effectiveness of FG-CCDB in mitigating spurious correlations, we analyze how sample reweighting influences the correlation between feature dimensions and class/bias information, as shown in Figure 3(left). We compute the correlation of each feature dimension with

class/bias information, and visualize their distributions using box plots Zhao et al. (2025). Before sample reweighting, strong spurious correlations in the training data lead the biased model $f_{\boldsymbol{\theta}_2}$ to rely heavily on bias-related features, with most dimensions exhibiting high correlation with bias and low correlation with class. After applying FG-CCDB weights on features from $f_{\boldsymbol{\theta}_2}$, the correlation with bias drops significantly, while the correlation with class increases. Moreover, after debiasing training on the reweighted data, this shift toward class-relevant features is further amplified, confirming that FG-CCDB effectively reduces the model's reliance on spurious features.

## 4.3 THE QUALITY OF BIAS EXPLORATION

In this section, we evaluate the effectiveness of MST by measuring its mode-prediction F1-score, precision, and recall against the ground-truth annotations. Results regarding the smallest-mode are shown in Figure4(b). Since our method progressively filters out bias-conflicting samples, it retains far fewer such samples than XRM and JTT, achieving the highest F1-score across the four datasets. This confirms the principle that removing bias-conflicting samples improves bias prediction. JTT misidentifies a large number of majority samples as belonging to the smallest-mode(low precision). In contrast, XRM tends to misidentify minority-modes samples as majority modes(low recall). With multi-stage refinement, our method achieving the best overall performance. As discussed in Section 3, repeating the "bias enhancement learning" process can further improve both bias prediction accuracy and consequently mode prediction accuracy. To validate this claim, we conduct experiments with different numbers of repetitions. The mode prediction performance across varying repetition counts are shown in Figure 4(a). The dashed lines represent the standard accuracy across all modes, while the solid lines show the recall for each individual mode. We observe that repetition has a particularly strong effect on minority groups (highlighted in bold), as evidenced by the significant improvement in their recall with more repetitions. Please refer to Appendix Figure 8 for convergence results with additional repetitions.

Figure 1(c) shows the final WGA for classification as repetition increases. Notably, performance improves substantially after the first repetition and then plateaus, especially on cMNIST and CelebA, suggesting that a single repetition is often sufficient to achieve satisfactory performance.

## 4.4 ABLATION STUDY

Our method comprises two core technical modules: MST and FG-CCDB, which together demonstrate superior performance. In this section, we integrate these modules with existing methods and observe the resulting performance improvements to verify the effectiveness and versatility of our approach, as detailed below.

($i$) To assess the effectiveness of MST, we replace ground-truth annotations in bias-supervised methods, *i.e.,* GroupDRO and DFR, with bias predictions generated by MST. This results in their unsupervised counterparts, denoted as GroupDRO-MST and DFR-MST, respectively. The results are reported at the top of Table 3. Remarkably, the generalization performance of these unsupervised variants is comparable to their supervised versions using human annotations. Although the bias predictions are not perfect, they are sufficiently accurate to identify most minority modes, confirming the effectiveness of our MST as an approximate substitute for human bias annotations.

($ii$) To evaluate the effectiveness of FG-CCDB independently of MST, we replace the predicted bias with human annotations, resulting in a supervised version, FG-CCDB-sup. The results are reported at the bottom of Table 3. When bias annotations are available, FG-CCDB-sup further boosts performance, achieving results comparable to existing supervised methods on Waterbirds,

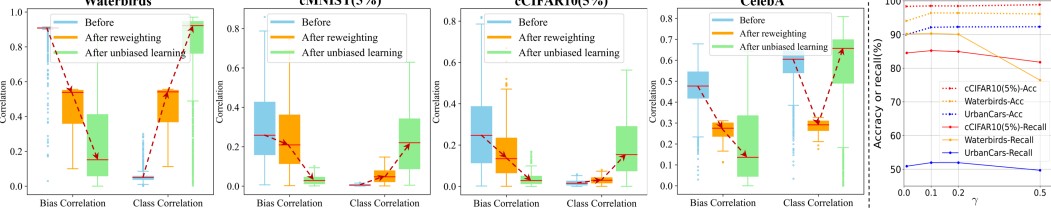

Figure 3: **Left:** effect of FG-CCDB sample reweighting in reshaping the data distribution and mitigating spurious correlations. **Right:** mode prediction accuracy and Smallest-mode recall along $\gamma$.

Figure 4: (a) The mode prediction accuracy along the repeating of the "bias enhancement learning" procedure; (b) Smallest-mode F1-score, precision, and recall compare with existing methods.

Table 5: The F1-score of the smallest-mode prediction under different top high-confidence ratio $\beta$.

| | cCIFAR10(5%) | Waterbirds | CelebA | UrbanCars |
|---|---|---|---|---|
| Bias-align ratio | 95.00% | 94.97% | 51.72% | 90.25% |
| $\beta = 30\%$ | 0.65 | 0.53 | 0.32 | 0.47 |
| $\beta = 50\%$ | 0.72 | 0.64 | **0.47** | 0.62 |
| $\beta = 70\%$ | **0.79** | **0.67** | 0.40 | **0.64** |

and outperforming them on the others, especially in multi-class settings. *This justified our statement on a more aggressive reweighting and indicates that FG-CCDB is a more effective strategy than naïve group balancing for handling spurious correlations.* Moreover, the performance gap between FG-CCDB-sup and the original FG-CCDB is marginal, further confirming the effectiveness of our method in reducing reliance on human bias annotations.

## 4.5 HYPERPARAMETER ANALYSIS

In this section, we evaluate the effect of the hyperparameters $\gamma$ for "Initial Bias Learning" and $\beta$ for selecting top high-confidence samples on MST's final performance. The results are presented in Figure 3(right) and Table 5.

The hyperparameter $\gamma$ controls the proportion of samples selected for training the initial bias model. Intuitively, a smaller $\gamma$ leads to stronger overfitting to bias cues and thus greater reliance on them. As expected, the results in Figure 3(right) show that when $\gamma \leq 0.2$, both prediction accuracy and smallest-mode recall remain high. However, as $\gamma$ increases to $0.5$, the performance drops significantly. We find that $\gamma = 0.1$ serves as a sweet spot, while also saving computation compared to $\gamma = 0.2$.

F1-score with $\beta \in \{30\%, 50\%, 70\%\}$ are reported in Table 5. The hyperparameter $\beta$ controls the proportion of top high-confidence samples selected to filter out bias-conflicting samples and amplify the model's bias. Intuitively, this value relates to the smallest bias-aligned ratio across classes, as shown in the first row of Table 5. Except for CelebA, whose ratio is slightly above $50\%$, all other datasets have ratios exceeding $90\%$. Accordingly, $\beta = 50\%$ serves as a reasonable middle-ground choice. For CelebA, which has a relatively low bias-aligned ratio, $\beta = 50\%$ achieves the best performance; whereas for datasets with ratios exceeding $90\%$, both $\beta = 50\%$ and $\beta = 70\%$ yield high F1-scores, with $\beta = 70\%$ performing the best. Intuitively, when bias annotations are unavailable, selecting the top $50\%$ high-confidence samples is likely to capture the bias-aligned subset while excluding bias-conflicting samples.

## 5 CONCLUSIONS

In this paper, we address the challenge of robust group generalization under spurious correlations without requiring bias annotations. Following the distribution matching paradigm, we propose a method that integrates a reliable bias prediction module with fine-grained class-conditional distribution matching. Our approach demonstrates strong performance on real-world datasets with single or multiple shortcuts, as well as highly biased multi-class datasets, often matching or outperforming methods that rely on human-provided group annotations. By leveraging the model's overfitting behavior, our method offers a novel alternative to traditional group balancing strategies and effectively reduces reliance on manual supervision. However, its effectiveness may be limited in scenarios where the overfitting signal fails to capture bias cues—for example, in CelebA, which has only one minority group, or in CivilComments, where majority groups dominate one class while minority groups appear in another. These settings present different spurious correlation patterns that weaken the overfitting signal used for bias prediction. Addressing this limitation remains an important direction for future research.

ACKNOWLEDGMENTS

This work was supported by the National Key R&D Program of China under Grant No. 2024YFA1012700. We thank Chenrong Li and Ruolan Liu for their assistance with figure preparation.

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
