Appendix for
Fine-Grained Class-Conditional Distribution Balancing for Debiased Learning

# A   THE ALGORITHM OF OUR PROPOSED METHOD

The complete procedure of our proposed FG-CCDB is summarized in Algorithm 1. It consists of three stages: bias exploration, sample weight inference, and unbiased classifier training.

---

**Algorithm 1** Fine-grained class-conditional distribution balancing (FG-CCDB)

---

**Input:** Randomly initialized network $f_{\boldsymbol{\theta}_1}$ and $f_{\boldsymbol{\theta}_2}$ for bias prediction, $f_{\boldsymbol{\phi}}$ for unbiased classification; training set $\mathcal{D}$, validation set $\mathcal{D}_v$.
**Output:** unbiased classifier $f_{\boldsymbol{\phi}}$.
*#Stage1: bias exploration via multi-stage data-selective retraining*
**1:** Randomly sample a subset $\mathcal{D}_1$ from $\mathcal{D}$ with proportion $\gamma$ ($\gamma = 10\%$).
**2:** Train $f_{\boldsymbol{\theta}_1}$ on $\mathcal{D}_1$ using ERM.
**3:** Select the top $\beta$ ($\beta = 50\%$) most biased samples from $\mathcal{D}$ to form an extremely biased subset $\mathcal{D}_1'$.
**4:** Train $f_{\boldsymbol{\theta}_2}$ on $\mathcal{D}_1'$ using ERM.
**5:** use $f_{\boldsymbol{\theta}_2}$ to infer bias labels for all samples in $\mathcal{D}$, and modeling joint distribution via hard confusion matrix.
*# Stage2: Sample weight inference*
**6:** Infer class-conditional and marginal distribution over the bias cues using equation2.
**7:** Compute sample weights using Equation3, and 4 from the main manuscript.
*# Stage 3: Unbiased classifier training*
**8:** Train classifier $f_{\boldsymbol{\phi}}$ on reweighted samples using standard ERM.
**9:** Select the best-performing $f_{\boldsymbol{\phi}}$ based on the highest worst-class accuracy on the validation set $\mathcal{D}_v$.

---

# B   EXPERIMENTAL SETUP

**Datasets.** The experiments are conducted on five benchmark datasets known to exhibit spurious correlations. Waterbirds, CelebA, CivilComments, and UrbanCars are real-world datasets in which each class is spuriously correlated with background, gender, certain demographic identities, or a combination of multiple shortcuts respectively. cMNIST and cCIFAR10 are synthetic ten-way classification tasks, where each class is spuriously linked to a specific color or noise pattern. For all datasets, we adopt the same train-validation-test split following Liu et al. (2021); Tsirigotis et al. (2023b) for fair comparison.

**Training setup.** For fair comparison, we adopt model architectures following Tsirigotis et al. (2023b); LaBonte et al. (2023b): a 3-hidden layer MLP for cMNIST, ResNet18 He et al. (2016) For cCIFAR10, ResNet50 He et al. (2016) for Waterbirds and CelebA, and BERT Devlin et al. (2019) for CivilComments. ResNet18 and ResNet50 are pretrained on ImageNet-1K, and BERT is pretrained on Book Corpus and English Wikipedia. No data augmentation is applied to cM-NIST and CivilComments, while simple augmentations (random cropping and horizontal flipping) are used to the remaining datasets, following Ahuja et al. (2021). This ensures that the improvements we observed are attributed to the proposed methodology, rather than to data augmentations that could potentially nullify the bias attribute. For our method, both the initial bias learning and the bias enhancement learning span 20 epochs, and the final unbiased learning involves 5000 iterations across all datasets. Results are averaged over 5 random seeds, and for each seed, the best-performing model (the one with the highest worst-class accuracy on the validation set Tsirigotis et al. (2023b)) is selected. Unless otherwise stated, we repeat the "bias enhancement learning" process three times for FG-CCDB.

**On Hyperparameters.** All experiments were conducted on a single NVIDIA A40 GPU. The hyperparameters and optimization settings for the MST and FG-CCDB modules on each dataset are summarized in Table 6. Both modules share the same batch size, scheduler, optimizer, and optimizer hyperparameters. For cMNIST and CivilComments, no data augmentation is applied to either

module, while for the remaining datasets, simple data augmentations (*i.e.,* ResizedCrop and HorizontalFlip) are applied only for the FG-CCDB module. All stages in MST are trained with the same *Epoch* number. In contrast to CCDB, our MST framework consists of at least two stages: the first stage provides an initial bias prediction, which is further refined by the subsequent stages. The experimental results in the main manuscript (see Figure3(right)) show that selecting the parameter $\gamma$ within the range of $1\% \leq \gamma \leq 20\%$ has a negligible impact on the final performance. Accordingly, we set $\gamma = 10\%$ across all datasets to ensure strong performance while maintaining low computational cost.

Table 6: The optimization setup for our FG-CCDB.

| Dataset | Optimizer | Scheduler | LR | Batch size | Weight decay | {Epoch,Iter} | $\gamma$ | Augmentation |
|---|---|---|---|---|---|---|---|---|
| cMNIST | Adam | None | $1 \times 10^{-2}$ | 256 | $1 \times 10^{-4}$ | $\{20, 5000\}$ | 0.1 | None |
| cCIFAR10 | Adam | None | $1 \times 10^{-5}$ | 256 | $1 \times 10^{-4}$ | $\{20, 5000\}$ | 0.1 | ResizedCrop, HorizontalFlip |
| Waterbirds | Adam | None | $1 \times 10^{-5}$ | 256 | $1 \times 10^{-4}$ | $\{20, 5000\}$ | 0.1 | ResizedCrop, HorizontalFlip |
| CelebA | Adam | None | $1 \times 10^{-5}$ | 256 | $1 \times 10^{-4}$ | $\{20, 5000\}$ | 0.1 | ResizedCrop, HorizontalFlip |
| CivilComments | AdamW | Linear | $1 \times 10^{-5}$ | 16 | $1 \times 10^{-4}$ | $\{20, 5000\}$ | 0.1 | None |

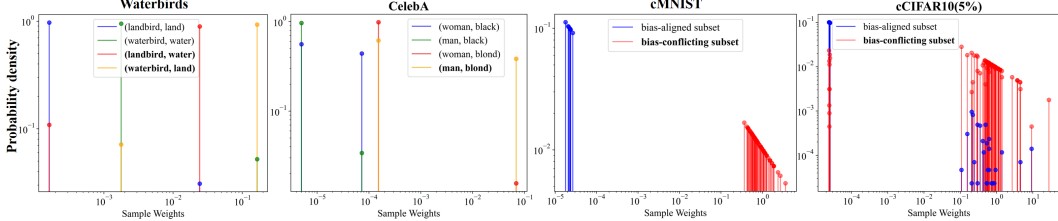

Figure 5: The distribution of the sample weights assigned by FG-CCDB within each mode on four datasets.

## C The sample weights inferred by our FG-CCDB

To assess whether our distribution-matching approach, FG-CCDB, effectively distinguishes minority modes from majority ones and assigns appropriate sample weights in the singular shortcut case, we analyze the distribution of inferred sample weights across different modes. The results on four datasets are summarized in Figure 5, with the minority modes highlighted in bold. As expected, samples from the majority modes are assigned low weights, typically concentrated below 0.01, while samples from minority modes receive significantly higher weights, clustered around 1. These results demonstrate that FG-CCDB successfully differentiates between majority and minority modes, and up-weights the latter in a balanced manner, aligning both class-conditional and marginal distributions.

## D Additional Experimental Results and Details for UrbanCars

For UrbanCars, the class label corresponds to the car type (country or urban), while the spurious attributes consist of two shortcuts: the background (BG) and the co-occurring object (CoObj), both of which are also labeled as country or urban. The ground-truth group partition of the training data is shown in Figure6. The majority groups contain urban car images combined with urban backgrounds (*e.g.,* alleys) and urban co-occurring objects (*e.g.,* fire plugs), and vice versa for country car images. The remaining combinations constitute the minority groups. As shown in Li et al. (2023b), mitigating spurious correlations in datasets with multiple shortcuts presents a Whac-A-Mole dilemma: mitigating one shortcut often amplifies the model's reliance on the others.

**Evaluation Metrics for the UrbanCars Dataset.** Compared to datasets with a single shortcut, four new metrics are proposed for multi-shortcut scenarios to better evaluate performance across different shortcut combinations.

Table 7: Classification performance on multi-shortcuts UrbanCars. In addition to our worst-group accuracy, the measurements following Li et al. (2023b) are also provided.

| Methods | Given Condition | I.D. Acc | Gap due to shortcut | | | Urbancar(BG) | | Urbancar(CoObj) | | Urbancar | |
|---|---|---|---|---|---|---|---|---|---|---|---|
| | | | BG | CoObj | BG+CoObj | Mean | WGA | Mean | WGA | Mean | WGA |
| LfF | Yes | 97.2 | -11.6 | -18.4 | -63.2 | - | - | - | - | - | - |
| JTT | Yes | 95.9 | -8.1 | -13.3 | -40.1 | - | - | - | - | - | - |
| DebiAN | No | 98.0 | -14.9 | -10.5 | -69.0 | - | - | - | - | - | - |
| ExMap | No | - | -5.9 | -9.9 | -30.7 | 93.2 | 71.4 | 93.2 | 79.2 | - | - |
| **FG-CCDB** | None | 92.98 | **-4.17** | **-7.37** | **-4.9** | $91.04_{\pm0.04}$ | $87.84_{\pm0.2.12}$ | $93.08_{\pm0.14}$ | $90.24_{\pm0.28}$ | $88.56_{\pm0.30}$ | $81.28_{\pm3.9}$ |

($i$) In-Distribution Accuracy (I.D. Acc): This metric computes the weighted average of per-group accuracies, where the weights are proportional to each group's frequency in the training set (*i.e.,* its correlation strength, as shown in Figure6). Following the "average accuracy" definition in Sagawa et al. (2020), it reflects model performance when no group shift occurs.

($ii$) BG Gap: The drop in accuracy from the I.D. Acc to the accuracy on groups where the background (BG) is uncommon but the co-occurring object (CoObj) remains common (*cf.* the first yellow column in Figure6).

($iii$) CoObj Gap: The drop in accuracy from the I.D. Acc to the accuracy on groups where the CoObj is uncommon but the BG remains common (*cf.* the second yellow column in Figure6).

($iv$) BG+CoObj Gap: The drop in accuracy from the I.D. Acc to the accuracy on groups where both BG and CoObj are uncommon (*cf.* the red column in Figure6).

BG Gap and CoObj Gap measure the model's robustness to distribution shifts caused by each individual shortcut. BG+CoObj Gap evaluates robustness in the most challenging scenario, where both shortcuts are absent.

Figure 6: Unbalanced groups in the UrbanCars training set based on two shortcuts: background and co-occurring object (the figure is adopted from Li et al. (2023b))

Following Chakraborty et al. (2024), two variants of UrbanCars are constructed: ($i$) UrbanCars (BG), where only the background object serves as the spurious attribute; ($ii$) UrbanCars (CoObj), where only the co-occurring object serves as the spurious attribute.

We compare the worst-group accuracy (WGA) on these two variants plus the original one, as shown in Table7. Our method achieves significantly higher WGA than ExMap on both variants, further confirming our claim that FG-CCDB captures bias information through mode partitioning in a more general manner. This makes it applicable to both singular and multiple shortcut scenarios.

# E  ADDITIONAL DISCUSSIONS

**R1W1: How iterative bias amplification improves minority-mode recall**

In addition to our experimental results, the validation of MST is supported by the following research findings: ($i$) Easy-to-learn property of bias attributes Nam et al. (2020). ERM tend to overfit spurious correlations only when they are "easier" to learn than the desired core features. This property has been successfully exploited in many debiasing methods Nam et al. (2020); Pezeshki et al. (2024); LaBonte et al. (2023b); Zhao et al. (2025); Lee et al. (2023) to detect and highlight underrepresented bias-conflicting samples. Thus, the initial step of MST is well motivated. ($ii$) Removing bias-

Table 8: The F1-score of the smallest-mode prediction under different top high-confidence ratio $\beta$.

|  | cCIFAR10(5%) | Waterbirds | CelebA | UrbanCars |
|---|---|---|---|---|
| Bias-align ratio | 95.00% | 94.97% | 51.72% | 90.25% |
| $\beta = 30\%$ | 0.65 | 0.53 | 0.32 | 0.47 |
| $\beta = 50\%$ | 0.72 | 0.64 | **0.47** | 0.62 |
| $\beta = 70\%$ | **0.79** | **0.67** | 0.40 | **0.64** |
| Adaptive | 0.76 | 0.69 | 0.43 | 0.66 |

conflicting samples improves bias prediction. Prior works Lee et al. (2023); Ciranni et al. (2025) show that even a small number of bias-conflicting samples can severely degrade the estimation of bias-aligned vs. bias-conflicting partitions. In principle, if all bias-conflicting samples were removed, one could train a bias-capturing model that provides ideal learning signals for debiasing. These methods obtain a bias-amplified model either by explicitly removing bias-conflicting samples or by generating only bias-aligning samples. Our MST shares the same core insight but adopts a different mechanism: we use a multi-stage bias amplification process that progressively filters out bias-conflicting samples by selecting those with the highest confidence. $(iii)$ Bias-aligned samples tend to have higher confidence. As revealed in Lee et al. (2023), bias attributes are easier to learn than intrinsic attributes; thus, ERM model assigns higher predicted probabilities to bias-aligned samples. This phenomenon has also been effectively used in works on GCE Zhang & Sabuncu (2018). Therefore, selecting top-confidence samples at each stage in MST is an effective strategy for filtering out bias-conflicting samples.

**R1Q1: Why fix the top-$50\%$ high-confidence samples per-class for bias enhancement?**

We denote by $\beta$ the ratio used to select the top high-confidence samples for brevity. Our choice of $\beta = 50\%$ is based on a practical and widely observed property of spurious-correlation datasets. In typical settings, within each class, the bias-aligned partition is larger than the bias-conflicting partition; otherwise, spurious correlations would not arise, as pointed out in Ciranni et al. (2025). This implies that the bias-aligned partition occupies more than $50\%$ of the samples in that class. Table 8 summarizes the smallest bias-aligned ratio across classes for each dataset. Except for CelebA, which has a value only slightly above $50\%$, the other datasets have ratios exceeding $90\%$. Therefore, when bias annotations are unavailable, selecting the top $50\%$ high-confidence samples is highly likely to capture the bias-aligned partition while excluding bias-conflicting samples. We emphasize that this is an empirical principle rather than a strict theoretical guarantee. However, it is consistently supported by prior works on spurious correlations and by our empirical results.

To further address potential concerns regarding $\beta = 50\%$, we conduct experiments with alternative proportions (30% and 70%) and an adaptive version based on class-wise confidence distributions (assigning higher $\beta$ to classes with higher average confidence). The F1-scores are shown in Table 8. Clearly, $\beta = 50\%$ represents a reasonable middle-ground option. For CelebA, which has a low bias-aligned ratio, $\beta = 50\%$ performs best, whereas for datasets with bias-aligned ratios above $90\%$, $\beta = 70\%$ yields the best performance. The adaptive strategy is primarily effective when the data exhibits noticeable class imbalance. We consider further exploration of this approach as promising future work.

**R2W1: How iterative bias amplification improves minority-mode recall**

Please refer to R1W1.

**R2W2: comparison with recent label-free debiasing methods**

We incorporate comparisons with recent label-free debiasing methods: DDB Ciranni et al. (2025), DaC Noohdani et al. (2024), and GERNE Asaad et al. (2025). DDB utilizes a diffusion model to generate bias-aligned data, which amplifies the bias reliance of the bias model and provides useful information for the debiasing process. DaC identifies the causal components of images using class activation maps from models trained with ERM. It then intervenes on the images by combining these components and retrains the model on the augmented data. Both DDB and DaC are specifically designed for image data. GERNE assumes that the difference between the gradients of two batches captures a debiasing direction and optimizes the model along this direction. The results are summarized in Table1, Table2 and Table4. Although DaC uses bias annotations during validation, its performance on CelebA remains significantly lower than ours. Our method demonstrates substantial advantages over GERNE and DDB across CelebA, CivilComments, and the multi-shortcut Urban-

Cars dataset. Notably, on UrbanCars, while DDB exhibits the smallest overall drops across different bias-conflicting scenarios, its base I.D. accuracy is the lowest among all compared methods.

**R2W3: The performance on multi-bias scenarios** The experiments on multi-bias (multi-shortcut) scenarios may have been overlooked. We conducted experiments on the UrbanCars dataset, which contains multiple shortcuts (i.e., background and co-occurring objects). The corresponding results and discussion can be found in Section 4.1 and Table 2.

Overall, our method consistently achieves the best balance between high I.D. accuracy and minimal drops compared to other bias-agnostic methods, particularly on the challenging BG+CoObj generalization. It performs comparably to — or better than — methods that rely on bias annotations. These results confirm that our approach provides a general framework for handling multi-shortcut scenarios.

**R2W4: whether FG-CCDB can compensate for imperfect bias predictions**

We have shown the performance of FG-CCDB under different mode partition qualities in Figure 1(c), which may have been overlooked. By observing Figure 4(a), we find that repetition has a particularly strong effect on minority groups: performance increases significantly after the first repetition and then gradually converges. Accordingly, in Figure 1(c), the WGA obtained by subsequent FG-CCDB shows a similar trend: it jumps from a relatively low accuracy after the first repetition and then gradually converges to a stable value. We conclude that: ($i$) When MST provides poor mode partitioning ("repeat0"), the errors are significant, and FG-CCDB is affected by these errors, resulting in relatively low WGA. ($ii$) When MST provides acceptable mode partitioning (with a repetition count of 1 or higher), the WGA of FG-CCDB increases and shows only marginal improvement with further repetitions, even though the mode partition quality continues to improve. This indicates that FG-CCDB can compensate for imperfect mode partitions once the partition quality is sufficiently high.

**R2Q1: how well the MST matches human labels? performance comparison results with the latest methods**

Please refer to R3W3 and R3W4 for a quantitative evaluation of MST's performance. Please refer to R2W2 for a comparison with the latest label-free and generative model-based methods.

**R3W1: Definition of 'mode' and whether major biases are captured by MST**

We define the "mode" $(s, y)$ as a black-box concept because our goal is for $s$ to capture general and harmful bias information that humans may not preconceive Li et al. (2022b), rather than only physically interpretable attributes. The value $s$ represents spurious signals that an ERM model prefers over core features and that consequently cause evaluation failures. We do not aim to model spurious attributes are not preferred by ERM and therefore do not lead to generalization errors. In this sense, model mistakes serve as indicators of harmful spurious correlations. Regarding the type of bias we focus on, we clarify that **our model is unlikely to fail to capture such harmful bias cues**. The reasons are as follows.

First, extensive prior works Nam et al. (2020); Pezeshki et al. (2024); LaBonte et al. (2023b); Zhao et al. (2025); Lee et al. (2023) operate under the widely accepted assumption that naive ERM tends to misclassify or produce low-confidence predictions on bias-conflicting samples. These studies demonstrate that ERM naturally learns spurious correlations, providing reliable learning signals for debiasing.

Second, for stronger theoretical grounding, we connect our idea to the Equal Opportunity Fairness (EOF) criterion Li et al. (2022b); Hardt et al. (2016) and show that our method is equivalent to find the bias cues that cause a classifier's predictions to strongly violate this fairness criterion, as detailed below.

Formally, a classifier $f$ satisfies EOF criterion if:

$$\Pr\{\hat{y} = k | s = 0, y = k\} = \Pr\{\hat{y} = k | s = 1, y = k\} \tag{5}$$

where the LHS and RHS are the true positive rates (TPR) for negative ($s = 0$) and positive ($s = 1$) groups in target class $k \in \{1...K\}$. As noted in Li et al. (2022b), a significant TPR discrepancy between groups indicates that classifier $f$ contains bias regarding $s$.

In our setting without bias annotations, we train an overfitted ERM and use its predictions $s$ as a general bias cues. Specifically, given a dataset $\mathcal{D}$ with spurious correlations, where minority groups are non-empty and target labels are correct, we train an ERM model $f$ on a small random subset of $\mathcal{D}$ and evaluate it on the full dataset, obtaining accuracy $a$.

- If $a = 100\%$, TPRs for each $(s, y)$ pair resemble Figure 7(a). This implies that bias cues are not preferred and $f$ likely relies exclusively on core features. No debiasing is needed.

- If $a < 100\%$, overfitting occurs, though to different degrees. The TPRs within each class show severe violations of the EOF criterion (*e.g.,* Figure 7(b) for class $k = 0$, $\Pr\{\hat{y} = 0|s = 0, y = 0\} \gg \Pr\{\hat{y} = 0|s \neq 0, y = 0\}$), indicating that $f$ indeed captures and relies on the bias encoded in $s$.

Thus, in principle, as long as $a < 100\%$, our method leveraging ERM overfitting reliably captures harmful implicit bias cues. Unlike Li et al. (2022b), our approach directly identifies cues that maximally violate EOF without requiring interleaving optimization.

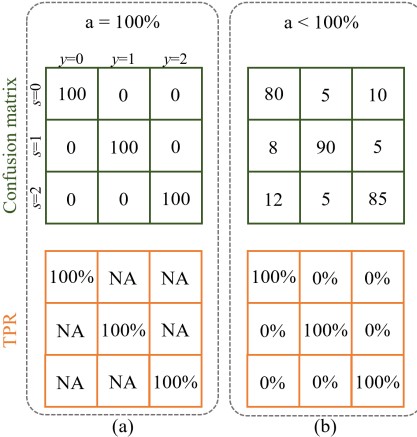

Figure 7: The hard confusion matrix and the TPRs. Take a 3-class classification task as an example, with each class contains 100 samples.

### R3W2: The hyperparameter choices in MST

In fact, we have conducted ablation studies on $\gamma$ in Figure 3(right) and discussed it in Section 4.4, which may have been overlooked. To further validate its robustness across datasets and bias strengths, we include additional results on UrbanCars. These results consistently show that $\gamma = 10\%$ serves as a sweet spot for maximizing the smallest-mode recall. Please refer to R1Q1 for our discussion regarding the use of the top $50\%$ high-confidence samples.

### R3W3: On MST's ability to capture complex biases in multi-shortcut scenarios

As we have pointed out in R3W1, we focus only on biases that are harmful — i.e., those that cause ERM models to overfit and make incorrect predictions — and our goal is to correct them. If the model overfits to "noise or irrelevant features" rather than physically interpretable biases, we treat such noise or irrelevant features as harmful bias and aim to balance them to improve ERM performance.

As demonstrated in Line 156 of the main manuscript, our model captures spurious cues that lead to overfitting and, consequently, incorrect predictions. These cues may correspond to interpretable shortcuts, combinations of multiple shortcuts, or entangled, uninterpretable patterns. Therefore, when multiple competing biases exist, MST can reveal the full bias structure, representing multiple competing biases within a single bias cue.

We have conducted experiments in Section 4.1 (Table 2) to demonstrate the effectiveness of our method in complex multi-shortcut scenarios, which may have been overlooked. For example, in UrbanCars, there are two competing shortcuts (background and co-occurring objects) and our method exhibits substantially less bias towards any specific background, co-object, or their combination, even outperforming methods that rely on multiple shortcut annotations.

Table 9: Recall of minority groups in UrbanCars predictions by MST, XRM, and JTT. Group $(e_1, e_2, e_3)$: $e_1 = 0/1$ indicates urban/country car, $e_2 = 0/1$ indicates urban/country object, and $e_3 = 0/1$ indicates urban/country background.

|      | (0,0,1) | (0,1,0) | (0,1,1) | (1,0,0) | (1,0,1) | (1,1,0) |
|------|---------|---------|---------|---------|---------|---------|
| MST  | **45.79%** | **58.42%** | **70.00%** | **100.00%** | **64.55%** | **28.57%** |
| XRM  | 41.05% | 30.51% | 0.00% | 60.00% | 10.12% | 14.06% |
| JTT  | 0.53% | 0.00% | 0.00% | 10.00% | 0.53% | 0.00% |

Additionally, we compare the Recall of bias-conflicting modes on UrbanCars obtained by XRM, JTT, and our MST in Table 9. The results show that even under multi-shortcut conditions, our method successfully identifies bias-conflicting samples covering all minority groups, whereas XRM fails to capture group $(0, 1, 1)$, and JTT fails to capture groups $(0, 1, 0)$, $(0, 1, 1)$, and $(1, 1, 0)$.

**R3W4: Quantitative Comparison of MST with XRM and DebiAN**.

The comparison with XRM and DebiAN on final debiasing performance was already provided in Table 1 and Table 2 of the main manuscript (Section 4.1), which may have been overlooked. Similarly, the comparison with XRM and JTT on bias capturing was already presented in Figure4(b) and discussed in section 4.3, which also may have been overlooked. Theoretically, XRM trains its biased model on a random half of the training data, which contains far more bias-conflicting samples than ours, resulting in lower precision and recall on the smallest-mode. A similar explanation accounts for JTT's poor recall. DebiAN uses an alternating training scheme, where the classifier gradually mitigates biases during the discovery phase, making it difficult for its discoverer to reliably predict biases; therefore, we do not include DebiAN in the bias-capturing comparison.

To provide a more comprehensive evaluation, we have additionally included the F1-score in Figure 4(b). Our method consistently achieves the highest F1-score.

**R3W5: F1-score to evaluate MST mode partitions**.

Please refer to R3W4 for quantitative evaluation of MST-generated mode partitions.

The effect of the MST's prediction quality on the subsequent FG-CCDB was already provided in Figure1(c) and may have been overlooked. Please refer to R2W4 for a detailed discussion.

**R3Q1: on further subdivision within modes or continuous weights**.

We appreciate the reviewer's insightful suggestion. While a mode may contain potential substructures, our assumption of intra-mode homogeneity is not a theoretical requirement but a practical approximation, motivated by the following: ($i$) the "mode" definition is conditioned on both the predicted bias and the label $(s, y)$. The auxiliary bias model partitions data according to the most dominant spurious patterns revealed by ERM overfitting. This ensures that samples assigned to the same mode share the most influential bias cues, which is sufficient for effective reweighting. In practice, these dominant bias cues account for the majority of generalization errors, while finer-grained variations within a mode have only marginal influence. ($ii$) Empirically, uniform per-mode weighting is stable and effective. We experimented with an alternative design (i.e., entropy-based intra-mode splitting that divides each mode into high-entropy and low-entropy subsets) but found it introduced noise and degrades performance. For example, on Waterbirds, WGA drops from $90.56\%$ to $89.90\%$, and on cCIFAR10 with an extremely small bias-conflicting portion (the smallest group contains only 19 samples), performance drops from $55.28\%$ to $50.18\%$. This suggests that finer intra-mode partitioning requires additional sub-bias cues to correctly guide matching, which are unavailable under the current setting.

FG-CCDB focuses on mode-level bias amplification guided by dominant shortcuts. Incorporating a more detailed internal structure is beyond the scope of this work. We therefore consider mode-level homogeneity a reasonable and empirically validated design trade-off, with finer-grained mode modeling left as future work.

**R3Q2: Performance curves over additional iterations to demonstrate MST convergence**.

We provide the mode partition and WGA results with additional repetition counts in Figure 8. The results show that when the number of repetitions exceeds 3, the improvement in mode-partition

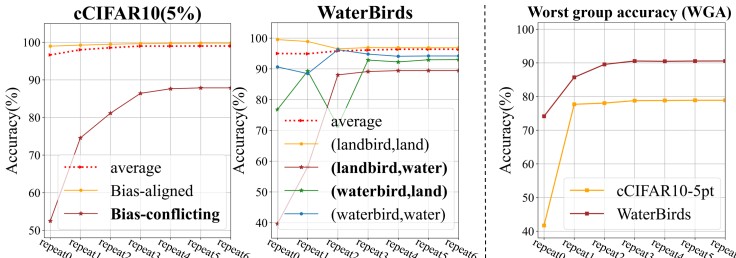

Figure 8: Mode prediction recall (left) and WGA under varying mode prediction quality (right) across repetitions of the "bias enhancement learning" procedure..

Table 10: The computation cost of compared methods on cCIFAR10.

|  | Our | ERM | uLA |
|---|---|---|---|
| Bias discovery | 80 epochs | NA | 500 epoch |
| Debiasing | 5000 iters≈28 epochs | 300 epoch | 500 epochs |

Table 11: The running time (hour) of MST, evaluated on a single NVIDIA A40 GPU.

|  | cCIFAR10 | Waterbirds | CelebA | UrbanCars |
|---|---|---|---|---|
| MST | 0.27h | 0.35h | 1.26h | 0.14h |

accuracy slows down and eventually converges to a stable point. Correspondingly, the WGA remains nearly unchanged once the repetition count is greater than 2.

This behavior is expected. As repetitions progresses, bias-conflicting samples are gradually filtered out, causing the bias-aligned ratio of the selected training subset to increase and eventually stabilize. Once the learned bias model reaches a stable level of bias reliance, further top-confidence selection no longer changes the bias-aligned ratio, and the mode partition consequently remains unchanged.

**R3Q3: Computational cost of MST**.

To avoid misunderstanding, uLA is also a two stage method. Compared to single-stage training methods like ERM, the additional training time mainly comes from MST. However, this overhead is acceptable in practical applications for the following reasons: ($i$) In each MST stage, we use only 10%, 50%, 50%, and 50% of the training data, which significantly reduces the computational burden. ($ii$) We observe that the ERM model already exhibits strong bias reliance in the early training phase — a phenomenon widely reported in prior works. Therefore, we set a small number of epochs for each MST stage. The main computation cost are compared in Table 10, and the running-time of MST is summarize in Table 11.