# OpenReview forum: "Fine-Grained Class-Conditional Distribution Balancing for Debiased Learning"
_ICLR.cc/2026/Conference — ICLR 2026 Poster_

### Official Review · Reviewer_FipK · 2025-10-29

**Soundness:** 2
**Presentation:** 3
**Contribution:** 2
**Rating:** 4
**Confidence:** 3

**Summary:**

This paper proposes a novel method named Fine-Grained Class-Conditional Distribution Balancing (FG-CCDB) to mitigate spurious correlations in deep learning models under the condition of no bias annotations. The authors point out that existing methods (e.g., CCDB) model class-conditional and marginal distributions as single Gaussian distributions, which are too coarse-grained to capture the complex multi-modal structures in real-world data. To address this, they design a **Multi-stage data-Selective reTraining (MST)** strategy that leverages model overfitting behavior to identify "modes" (i.e., bias patterns) through multi-stage data filtering. Based on this, FG-CCDB performs distribution alignment at the mode level, achieving more precise sample reweighting. Experimental results show that the method can rival bias-supervised approaches in binary classification tasks and even outperform them in multi-class tasks, with low computational overhead.

**Strengths:**

（1）Proposed Multi-stage data-Selective reTraining (MST), which utilizes model overfitting to construct highly biased training sets through multi-stage data filtering. This generates reliable pseudo-bias labels and forms a hard confusion matrix to characterize fine-grained "mode" structures, offering a new perspective for unsupervised bias exploration.
（2）Proposed FG-CCDB, which performs class-conditional and marginal distribution alignment at the mode level based on MST. This achieves finer and more effective distribution matching than the original CCDB, effectively mitigating spurious correlations.
（3）The method requires no ground-truth bias annotations, is computationally efficient, and allows sample weights to be computed via closed-form solutions. It achieves superior or comparable performance to existing unsupervised and partially supervised methods on multiple binary and multi-class benchmarks, particularly in strong-bias multi-class scenarios.

**Weaknesses:**

（1）The definition of "mode" is dependent on specific bias model architectures and training processes, lacking semantic clarity and theoretical guarantees.
The authors define a mode as (s, y), where s is a "pseudo-bias label" predicted by an auxiliary bias model. However, the physical meaning of s is ambiguous—it may correspond to a single shortcut, a combination of shortcuts, or even entangled uninterpretable patterns. This makes "mode" a black-box concept, with unclear mapping to latent bias factors in the data generation process. Moreover, the quality of s heavily relies on the initial ERM model's overfitting behavior. If the model fails to capture major biases, the entire MST process may collapse.

（2）Hyperparameter choices in MST (e.g., γ=10%, top 50% high-confidence samples) lack systematic analysis and generalization guarantees.
While the authors claim γ=10% is a "sweet spot," this conclusion seems empirical, with no sufficient ablation studies to validate its robustness across datasets or bias strengths. Similarly, selecting the top 50% high-confidence samples appears arbitrary, with no discussion of alternative proportions (e.g., 30% or 70%). These critical hyperparameters lack theoretical justification or adaptive mechanisms, reducing the method's universality and reproducibility.

（3）The method's validity critically depends on the strong assumption that "overfitted models accurately reflect bias structures," which may not hold in complex real-world scenarios.
The core idea is to exploit ERM models' overfitting to reveal biases. However, in real-world data, biases may be subtle or diverse, and models might overfit to noise or irrelevant features instead of true bias cues. Additionally, when multiple competing biases exist, the model may capture only one, leaving MST unable to reveal the full bias structure. While experiments show strong performance, the robustness of this assumption has not been validated in more challenging scenarios with hidden or complex biases.

（4）Insufficient comparison with existing methods (e.g., XRM, DebiAN) to highlight FG-CCDB's core innovations.
The paper includes XRM and DebiAN as baselines but provides brief descriptions of their mechanisms. XRM uses auxiliary models to generate pseudo-group labels, while DebiAN iteratively trains a "discoverer" model to identify biases. MST is essentially another pseudo-labeling strategy. Reviewers expect a more detailed comparison: e.g., does MST's pseudo-label quality significantly outperform XRM or DebiAN? On which bias types does MST excel? Otherwise, FG-CCDB may appear as another variant of existing pseudo-labeling + reweighting paradigms rather than a fundamental breakthrough.

（5）The experimental evaluation lacks direct quantitative analysis of mode partitioning quality.
While Figure 2(b) shows a heatmap of the joint distribution J and claims it reflects true biases, there are no quantitative metrics (e.g., mutual information, F1-score with ground-truth biases) to evaluate the accuracy of MST-generated mode partitions. If the mode partitioning itself contains significant errors, subsequent FG-CCDB distribution matching may amplify these errors, leading to performance degradation. The authors claim MST "serves as a strong proxy for ground-truth bias annotations," but this assertion requires stronger empirical or theoretical evidence.

**Questions:**

（1）FG-CCDB assumes samples within a mode are homogeneous and assigns them equal weights. However, a mode (e.g., a specific background for a class) may still have substructures or diversity. Does ignoring intra-mode heterogeneity affect reweighting effectiveness? Have you considered further subdivision within modes or introducing continuous weights?

（2）In multi-stage MST, repeating "bias enhancement learning" up to three times improves performance. Is there a saturation point? Would further iterations cause model collapse or overfitting to noise? Could you provide performance curves across iteration counts to demonstrate convergence?

（3）Although FG-CCDB is computationally efficient, MST requires training multiple auxiliary models, involving multiple training stages. Compared to single-training methods like ERM or uLA, how much additional training time does it incur? Is this computational cost acceptable in practical applications?

---

> ### Author Response · Authors · 2025-11-21
>
> **R3W1**: Definition of ‘mode’ and whether major biases are captured by MST
>
> We define the "mode" $(s,y)$ as a black-box concept because our goal is for $s$ to capture general and harmful bias information that humans may not preconceive [1], rather than only physically interpretable attributes. The value $s$ represents spurious signals that an ERM model prefers over core features and that consequently cause evaluation failures. We do not aim to model spurious attributes are not preferred by ERM and therefore do not lead to generalization errors. In this sense, model mistakes serve as indicators of harmful spurious correlations.
> Regarding the type of bias we focus on, we clarify that **our model is unlikely to fail to capture such harmful bias cues**. The reasons are as follows.
>
> ($i$) Extensive prior works operate under the widely accepted assumption that naive ERM tends to misclassify or produce low-confidence predictions on bias-conflicting samples [7]. These studies demonstrate that ERM naturally learns spurious correlations, providing reliable learning signals for debiasing.
>
> ($ii$)  For stronger theoretical grounding, we connect our idea to the Equal Opportunity Fairness (EOF) criterion [1] and show that our method is equivalent to find the bias cues that cause a classifier’s predictions to strongly violate this fairness criterion, as detailed below (please refer to the **appendix section E, R3W1 and Figure 7** for a detailed demonstration).
>
> Formally, a two-way classifier $f$ satisfies EOF criterion if:
> $$\text{Pr}  ( \hat{y}=k|s=0,y=k )  = \text{Pr} ( \hat{y}=k|s=1,y=k )$$
> where the LHS and RHS are the true positive rates (TPR) for negative ($s=0$) and positive ($s=1$) groups in target class $k\in \{1...K\}$.
> **As noted in [1], a significant TPR discrepancy between groups indicates that classifier $f$ contains bias regarding $s$.**
>
> In our setting without bias annotations, we train an overfitted ERM and use its predictions $s$ as a general bias cues.
> Specifically, given a dataset $\mathcal{D}$ with spurious correlations, where minority groups are non-empty and target labels are correct, we train an ERM model $f$ on a small random subset of $\mathcal{D}$ and evaluate it on the full dataset, obtaining accuracy $a$.
>
> - If $a=$100%, TPRs for each $(s,y)$ pair resemble Figure 7(a). This implies that bias cues are not preferred and $f$ likely relies exclusively on core features. No debiasing is needed.
> - If $a<$100%, overfitting occurs, though possibly to varying degrees. The TPRs within each class show severe violations of the EOF criterion (e.g., Figure 7(b) for class $k=0$, where $\text{Pr}  ( \hat{y}=0|s=0,y=0 ) \gg\text{Pr}  ( \hat{y}=0|s\neq 0,y=0 ) $), indicating that $f$ indeed captures and relies on the bias encoded in $s$.
>
> Thus, **in principle, as long as overfit occurs  (classification accuracy below 100%), our method leveraging ERM overfitting reliably captures harmful implicit bias cues.**
> Unlike [1], our approach directly identifies cues that maximally violate EOF without requiring interleaving optimization.
>
> **R3W2**: The hyperparameter choices in MST
>
> In fact, we conducted ablation studies on $\gamma$ (Figure 3, right), which corresponds to Figure 4(c) in the previous submission, and discussed the results in Section 4.5 of the main manuscript, corresponding to Section 4.4 in the previous submission. These points may have been overlooked.
>
> To further validate its robustness across datasets and bias strengths, we include additional results on UrbanCars in Figure 3(right).
> These results consistently show that $\gamma=$10% serves as a sweet spot for maximizing the smallest-mode recall.
>
> Please refer to R1Q1 for our discussion regarding the use of the top 50% high-confidence samples.
>
> [1] Zhiheng Li, Anthony Hoogs, and Chenliang Xu. Discover and mitigate unknown biases with debiasing alternate networks. In European Conference on Computer Vision, pp. 270–288. Springer.
> [7] Tyler LaBonte, Vidya Muthukumar, and Abhishek Kumar. Towards last-layer retraining for group robustness with fewer annotations. NeurIPS, 36:11552–11579, 2023.

---

> ### Author Response · Authors · 2025-11-21
>
> **R3W3**: On MST’s ability to capture complex biases in multi-shortcut scenarios.
>
> As we pointed out in R3W1, we focus only on biases that are harmful --- i.e., those that cause ERM models to overfit and make incorrect predictions --- and our goal is to correct them.
> If the model overfits to “noise or irrelevant features” rather than physically interpretable biases, we treat such “noise or irrelevant features” as harmful bias and aim to balance them to improve ERM performance.
>
> As demonstrated in Line 156-172 of the main manuscript, our model captures spurious cues that lead to overfitting and, consequently, incorrect predictions. These cues may correspond to interpretable shortcuts, combinations of multiple shortcuts, or entangled, uninterpretable patterns. **Therefore, when multiple competing biases exist, MST can reveal the full bias structure, representing multiple competing biases within a single bias cue.**
>
> We have conducted experiments in Section 4.1 (Table 2) to demonstrate the effectiveness of our method in complex multi-shortcut scenarios, which may have been overlooked. For example, in UrbanCars, there are two competing shortcuts (background and co-occurring objects) and our method exhibits substantially less bias towards any specific background, co-object, or their combination, even outperforming methods that rely on multiple shortcut annotations.
>
> Additionally, we compare the Recall of bias-conflicting modes on UrbanCars obtained by XRM, JTT, and our MST in the following table (also Table 9 in the appendix). The results show that even under multi-shortcut conditions, our method successfully identifies bias-conflicting samples covering all minority groups, whereas XRM fails to capture group $(0,1,1)$, and JTT fails to capture groups $(0,1,0)$, $(0,1,1)$, and $(1,1,0)$.
>
> | |(0,0,1) | (0,1,0) | (0,1,1) | (1,0,0) | (1,0,1) |(1,1,0) |
> | ------- | ------- | ------- | ------- | ------- | ------- | ------- |
> | MST |45.79%|58.42%|70.00%|100.00%|64.55%|28.57%  |
> | XRM | 41.05% | 30.51% | 0.00% | 60.00% |10.12% | 14.06% |
> |JTT | 0.53% | 0.00% | 0.00% | 10.00% | 0.53% | 0.00%|
>
> **R3W4**: Quantitative Comparison of MST and FG-CCDB with XRM and DebiAN
>
> The comparison of FG-CCDB with XRM and DebiAN on final debiasing performance was already provided in Table 1 and Table 2 of the main manuscript (Section 4.1), which may have been overlooked.
> Similarly, the comparison of MST with XRM and JTT on bias capturing was already presented in Figure4(b) and discussed in section 4.3, which also may have been overlooked.
> Theoretically, XRM trains its biased model on a random half of the training data, which contains far more bias-conflicting samples than ours, resulting in lower precision and recall on the smallest-mode. A similar explanation accounts for JTT's poor recall. DebiAN uses an alternating training scheme, where the classifier gradually mitigates biases during the discovery phase, making it difficult for its discoverer to reliably predict biases; therefore, we do not include DebiAN in the bias-capturing comparison.
>
> To provide a more comprehensive evaluation, we have additionally included the F1-score in Figure 4(b). Our method consistently achieves the highest F1-score.
>
> **R3W5**: F1-score to evaluate MST mode partitions; the effect of the MST's prediction quality on the subsequent FG-CCDB?
>
> Please refer to R3W3 and R3W4 for quantitative evaluation of MST-generated mode partitions.
> The effect of the MST's prediction quality on the subsequent FG-CCDB was already provided in Figure1(c) and may have been overlooked. Please refer to R2W4 for additional discussion.

---

> ### Author Response · Authors · 2025-11-21
>
> **R3Q1**: on further subdivision within modes or continuous weights
>
> We appreciate the reviewer’s insightful suggestion. While a mode may contain potential substructures, our assumption of intra-mode homogeneity is not a theoretical requirement but a practical approximation, motivated by the following:
> - The "mode" definition is conditioned on both the predicted bias and the label $(s,y)$.
> The auxiliary bias model partitions data according to the most dominant spurious patterns revealed by ERM overfitting. This ensures that samples assigned to the same mode share the most influential bias cues, which is sufficient for effective reweighting. In practice, these dominant bias cues account for the majority of generalization errors, while finer-grained variations within a mode have only marginal influence.
> - Empirically, uniform per-mode weighting is stable and effective. We experimented with an alternative design (i.e., entropy-based intra-mode splitting that divides each mode into high-entropy and low-entropy subsets) but found it introduced noise and degrades performance.
> For example, on Waterbirds, WGA drops from 90.56% to 89.90%, and on cCIFAR10 with an extremely small bias-conflicting portion (the smallest group contains only 19 samples), performance drops from 55.28% to 50.18%.
> This suggests that finer intra-mode partitioning requires additional sub-bias cues to correctly guide matching, which are unavailable under the current setting.
>
> FG-CCDB focuses on mode-level bias mitigation guided by dominant shortcuts. Incorporating a more detailed internal structure is beyond the scope of this work. We therefore consider mode-level homogeneity a reasonable and empirically validated design trade-off, with finer-grained mode modeling left as future work.
>
> **R3Q2**: Performance curves over additional iterations to demonstrate MST's convergence
>
> We provide the mode partition and WGA results with additional repetition counts in Figure 8 of the appendix.
> The results show that when the number of repetitions exceeds 3, the improvement in mode-partition accuracy slows down and eventually converges to a stable point (Figure 8 (left)). Correspondingly, the WGA remains nearly unchanged once the repetition count is greater than 2 (Figure 8 (right)).
>
> This behavior is expected. As repetitions progresses, bias-conflicting samples are gradually filtered out, causing the bias-aligned ratio of the selected training subset to increase and eventually stabilize. Once the learned bias model reaches a stable level of bias reliance, further top-confidence selection no longer changes the bias-aligned ratio, and the mode partition consequently remains unchanged.
>
> **R3Q3**: Computational cost of MST.
>
> To avoid misunderstanding, uLA is also a two stage method.
> Compared to single-stage training methods like ERM, the additional training time of our method mainly comes from MST. However, this overhead is acceptable in practical applications for the following reasons:
> - In each MST stage, we use only 10%, 50%, 50%, and 50% of the training data, which significantly reduces the computational burden.
> - We observe that the ERM model already exhibits strong bias reliance in the early training phase --- a phenomenon widely reported in prior works. Therefore, we set a small number of epochs for each MST stage.
>
> The following results also shows that the computational cost of MST is acceptable.
> - The main computation cost are compared in the following table (also Table 10 of the appendix, using cCIFAR10 (5%) as an example).
>
> | |Our | ERM | uLA |
> | ------- | ------- | ------- | ------- |
> | Bias discovery | 80 epochs   | NA | 500 epoch |
> | Debiasing| 5000 iters$\approx$28 epochs | 300 epoch | 500 epochs |
>
> - The running-time (hour) of MST is summarize in the following table (also Table 11 of the appendix, using a single NVIDIA A40 GPU).
>
> | |cCIFAR10 | Waterbirds | CelebA | UrbanCars|
> | ------- | ------- | ------- | ------- | ------- |
> | MST | 0.27h | 0.35h | 1.26h | 0.14h |

---

### Official Review · Reviewer_exPH · 2025-10-30

**Soundness:** 2
**Presentation:** 3
**Contribution:** 2
**Rating:** 6
**Confidence:** 4

**Summary:**

This paper builds upon the existing Class-Conditional Distribution Balancing (CCDB) framework and proposes two key components:
- MST (Multi-stage data-Selective reTraining): a strategy to characterize the bias structure through the hard confusion matrix, serving as a proxy for bias annotations.
- FG-CCDB (Fine-Grained Class-Conditional Distribution Balancing) - a fine-grained extension of CCDB that performs distribution alignment at the mode level, enabling a more detailed representation of multi-modal data distributions.

The method aims to achieve annotation-free debiasing and robust generalization by modeling complex intra-class variations caused by spurious correlations. Experimental results show that MST can substitute for bias annotations in supervised baselines (e.g., GroupDRO, DFR), and FG-CCDB outperforms prior approaches in multi-class, bias-heavy scenarios.

**Strengths:**

- Well-structured and clearly written:
The paper is logically organized, and the presentation of ideas—from motivation to methodological formulation—is easy to follow.

- Conceptual improvement over CCDB:
By replacing the single-Gaussian assumption with a multi-modal, mode-based distribution matching framework, the paper effectively extends CCDB to more realistic data distributions.

- Novel use of confusion matrix:
Employing the confusion matrix to infer bias-aligned and bias-conflicting “modes” is elegant and intuitively appealing.
It offers a discrete, bias-agnostic way to describe intra-class spurious correlations.

- Annotation-free contribution:
The approach is promising in scenarios where human bias annotations are unavailable or infeasible, showing comparable results to bias-supervised baselines.

**Weaknesses:**

- Indirect validation of MST as bias substitute:
The core assumption that MST can replace human-provided bias annotations is only indirectly validated through final task performance. The paper does not report any direct quantitative measure (e.g., F1, ARI, or NMI) of how well MST’s predicted bias partitions align with human-labeled bias groups. Without this, it is unclear whether MST truly captures bias structure or simply produces partitions that happen to improve performance.

- Limited comparison with recent label-free debiasing methods:
Although the paper positions itself within the “annotation-free” literature, it lacks comparisons with the latest label-free or label-free-from-features (LFF) methods, such as those that employ generative modeling or causal data augmentation for debiasing. Including these would strengthen the empirical validity and demonstrate broader applicability.

- Experimental generality:
Most experiments are conducted on benchmark datasets (e.g., Waterbirds, CMNIST) with relatively simple, well-defined bias sources.
The performance of MST and FG-CCDB under multi-bias or entangled bias scenarios remains uncertain.

- Ablation analysis depth:
While ablations for MST and FG-CCDB are presented, the interaction between the two modules is not deeply analyzed. It is unclear how errors from MST propagate to FG-CCDB, or whether FG-CCDB can compensate for imperfect bias predictions.

**Questions:**

As the paper claims, can the authors experimentally demonstrate how well the MST matches human labels?
Can the authors of the paper present performance comparison results with the latest methods using 'other label free + generative model methods'?

---

> ### Author Response · Authors · 2025-11-21
>
> **R2W1**: How iterative bias amplification improves minority-mode recall
>
> Please refer to R1W1.
>
> **R2W2**: comparison with recent label-free debiasing methods
>
> We incorporate comparisons with recent label-free debiasing methods: DDB [2], DaC [3], and GERNE [4].
> DDB utilizes a diffusion model to generate bias-aligned data, which amplifies the bias reliance of the bias model and provides useful information for the debiasing process.
> DaC identifies the causal components of images using class activation maps from models trained with ERM. It then intervenes on the images by combining these components and retrains the model on the augmented data.
> Both DDB and DaC are specifically designed for image data.
> GERNE assumes that the difference between the gradients of two batches captures a debiasing direction and optimizes the model along this direction.
> The results are summarized in Table1, Table2 and Table4.
>
> Although DaC uses bias annotations during validation, its performance on CelebA remains significantly lower than ours. Our method demonstrates substantial advantages over GERNE and DDB across CelebA, CivilComments, and the multi-shortcut UrbanCars dataset. Notably, on UrbanCars, while DDB exhibits the smallest overall drops (compared to I.D. accuracy) across different bias-conflicting scenarios, its base I.D. accuracy is the lowest among all compared methods.
>
> **R2W3**: The performance on multi-bias scenarios
>
> The experiments on multi-bias (multi-shortcut) scenarios may have been overlooked. We conducted experiments on the UrbanCars dataset, which contains multiple shortcuts (i.e., background and co-occurring objects). The corresponding results and discussion can be found in Section 4.1 and Table 2.
>
> Overall, our method consistently achieves the best balance between high I.D. accuracy and minimal drops compared to other bias-agnostic methods, particularly on the challenging BG+CoObj generalization. It performs comparably to --- or better than --- methods that rely on bias annotations.
> These results confirm that our approach provides a general framework for handling multi-shortcut scenarios.
>
> **R2W4**: whether FG-CCDB can compensate for imperfect bias predictions
>
> We have shown the performance of FG-CCDB under different mode partition qualities in Figure 1(c) of the main manuscript, which may have been overlooked.
> By observing Figure 4(a) of the main manuscript, we find that repetition has a particularly strong effect on minority groups: performance increases significantly after the first repetition and then gradually converges.
> Accordingly, in Figure 1(c), the WGA obtained by subsequent FG-CCDB shows a similar trend: it jumps from a relatively low accuracy after the first repetition and then gradually converges to a stable value.
> We conclude that:
> - When MST provides poor mode partitioning ("repeat0"), the errors are significant, and FG-CCDB is affected by these errors, resulting in relatively low WGA.
> - When MST provides acceptable mode partitioning (with a repetition count of 1 or higher), the WGA of FG-CCDB increases and shows only marginal improvement with further repetitions, even though the mode partition quality continues to improve. This indicates that FG-CCDB can compensate for imperfect mode partitions once the partition quality is sufficiently high.
>
> **R2Q1**: how well the MST matches human labels? performance comparison results with the latest methods
>
> Please refer to R3W3 and R3W4 for a quantitative evaluation of MST's performance.
> Please refer to R2W2 for a comparison with the latest label-free and generative model-based methods.
>
> [2] Massimiliano Ciranni, Vito Paolo Pastore, Roberto Di Via, Enzo Tartaglione, Francesca Odone, and Vittorio Murino. Diffusing debias: Synthetic bias amplification for model debiasing. In NeurIPS, 2025.
> [3] Fahimeh Hosseini Noohdani, Parsa Hosseini, Aryan Yazdan Parast, Hamidreza Yaghoubi Araghi, and Mahdieh Soleymani Baghshah. Decompose-and-compose: A compositional approach to mitigating spurious correlation. In CVPR, pp. 27662–27671, June 2024.
> [4] Ihab Asaad, Maha Shadaydeh, and Joachim Denzler. Gradient extrapolation for debiased representation learning. In ICCV, 2025.

---

> > ### Comment · Reviewer_exPH · 2025-11-27
> >
> > Thank you for providing a thorough rebuttal.
> > I have carefully reviewed your responses to my comments, particularly your clarification on a quantitative evaluation of MST's performance.
> >
> > While I appreciate the effort and the additional information provided, my fundamental concerns regarding a soundness of your work was not fully alleviated.
> >
> > Therefore, I have decided to maintain my original score of 6.
> > Thanks.

---

### Official Review · Reviewer_2Y79 · 2025-10-31

**Soundness:** 3
**Presentation:** 2
**Contribution:** 3
**Rating:** 6
**Confidence:** 3

**Summary:**

This paper addresses debiased learning under spurious correlations without relying on bias annotations. It targets the problem that existing methods mitigate spurious correlations by matching class-conditional to marginal distributions but rely on overly coarse single-Gaussian approximations that fail in multi-modal, multi-class settings. The authors propose MST to exploit ERM overfitting, training on a small split, and amplifying bias via per-class top-confidence selection to derive discrete modes from the final confusion matrix. Then they propose FG-CCDB to transform this matrix into mode-wise weights that align class-conditional to marginal distributions, reducing spurious reliance with lightweight, scalable computation. Experiments that MST can effectively substitute for human bias annotations in supervised methods, and FG-CCDB consistently outperforms or matches state-of-the-art bias-agnostic baselines while maintaining low computational and memory overhead.

**Strengths:**

-The paper offers a thorough and compelling analysis of the limitations in prior work, clearly diagnosing CCDB’s single-Gaussian assumption and proposing an elegant, scalable remedy via mode-wise matching derived from confusion-matrix–based distributions.
-The approach demonstrates strong practical value, showing robustness to multi-shortcut scenarios (e.g., UrbanCars) with competitive or superior performance relative to both bias-agnostic and supervised baselines.
-The experimental evaluation is comprehensive. Ablations cleanly disentangle the contributions of MST and FG-CCDB, and the correlation-shift analyses substantiate the mechanism that the method reduces reliance on bias-related features.

**Weaknesses:**

-The proposed mechanism lacks theoretical grounding. There is no formal analysis of how iterative bias amplification improves minority-mode recall. While the empirical evidence is compelling, a theoretical treatment would substantially strengthen the contribution.
-Several methodological details require further clarification. The selection of top-confidence samples hinges on the biased model’s calibration. Miscalibration could distort mode discovery, yet no temperature scaling or calibration baseline is reported.

**Questions:**

Why fix the top-50% per-class in bias enhancement? Did you explore adaptive thresholds (e.g., based on class-wise confidence distributions, entropy) or rank-based schedules across iterations? Any results on temperature scaling to improve confidence reliability?

---

> ### Author Response · Authors · 2025-11-21
>
> **R1W1** : How iterative bias amplification improves minority-mode recall
>
> In addition to our experimental results, the validation of MST is supported by the following research findings:
>
> ($i$) Easy-to-learn property of bias attributes [5]. ERM tend to overfit spurious correlations only when they are "easier" to learn than the desired core features.
> This property has been successfully exploited in many debiasing methods to detect and highlight underrepresented bias-conflicting samples.
> Thus, the initial step of MST is well motivated.
>
> ($ii$) Removing bias-conflicting samples improves bias prediction. Prior works [2,6] show that even a small number of bias-conflicting samples can severely degrade the estimation of bias-aligned vs. bias-conflicting partitions. In principle, if all bias-conflicting samples were removed, one could train a bias-capturing model that provides ideal learning signals for debiasing. These methods obtain a bias-amplified model either by explicitly removing bias-conflicting samples or by generating only bias-aligning samples.
> Our MST shares the same core insight but adopts a different mechanism: we use a multi-stage bias amplification process that progressively filters out bias-conflicting samples by selecting those with the highest confidence.
>
> ($iii$) Bias-aligned samples tend to have higher confidence. As revealed in [6], bias attributes are easier to learn than intrinsic attributes; thus, ERM model assigns higher predicted probabilities to bias-aligned samples. This phenomenon has also been effectively used in works on GCE loss [8].
> Therefore, selecting top-confidence samples at each stage in MST is an effective strategy for filtering out bias-conflicting samples.
>
> These points together provide strong support for the design and validation of MST.
>
> **R1Q1**: Why fix the top-50% high-confidence samples per-class for bias enhancement?
>
> We denote by $\beta$ the ratio used to select the top high-confidence samples for brevity.
> Our choice of $\beta=$50% is based on a practical and widely observed property of spurious-correlation datasets. In typical settings, within each class, the bias-aligned partition is larger than the bias-conflicting partition; otherwise, spurious correlations would not arise, as pointed out in [2]. This implies that the bias-aligned partition occupies more than 50% of the samples in that class.
> The following table  (also Table 8 in the appendix) summarizes the smallest bias-aligned ratio across classes for each dataset. Except for CelebA, which has a value only slightly above 50%, the other datasets have ratios exceeding 90%.
> Therefore, when bias annotations are unavailable, selecting the top 50% high-confidence samples is highly likely to capture the bias-aligned partition while excluding bias-conflicting samples. We emphasize that this is an empirical principle rather than a strict theoretical guarantee. However, it is consistently supported by prior works on spurious correlations and by our empirical results.
>
> To further address potential concerns regarding $\beta=$50%, we conduct experiments with alternative proportions (30% and 70%) and an adaptive version based on class-wise confidence distributions (assigning higher $\beta$ to classes with higher average confidence). The F1-scores are shown in the table below (also Table 8 in the appendix). Clearly, $\beta=$50% represents a reasonable middle-ground option.
> For CelebA, which has a low bias-aligned ratio, $\beta=$50% performs best, whereas for datasets with bias-aligned ratios exceeding 90%, both $\beta = $50% and $\beta = $70% yield high F1-scores, with $\beta = $70% performing the best.
> The adaptive strategy is primarily effective when the data exhibits noticeable class imbalance. We consider further exploration of this approach as promising future work. A detailed analysis has been revised in Section 4.5.
>
> |   | cCIFAR10(5%) | Waterbirds | CelebA | UrbanCars  |
> | ------- | ------- | ------- | ------- | ------- |
> | Bias-align ratio | 95.00% | 94.97% | 51.72% | 90.25% |
> | $\beta$=30%| 0.65 | 0.53 | 0.32 | 0.47 |
> | $\beta$=50% | 0.72 | 0.64 | 0.47 | 0.62|
> |$\beta$=70% | 0.79|0.67| 0.40 |0.64|
> |Adaptive |0.76 | 0.69 | 0.43 | 0.66|
>
> [2] Massimiliano Ciranni, Vito Paolo Pastore, Roberto Di Via, Enzo Tartaglione, Francesca Odone, and Vittorio Murino. Diffusing debias: Synthetic bias amplification for model debiasing. NeurIPS, 2025.
> [5] Junhyun Nam, Hyuntak Cha, Sungsoo Ahn, Jaeho Lee, and Jinwoo Shin. Learning from failure:
> De-biasing classifier from biased classifier. NeurIPS,33:20673–20684, 2020.
> [6] Jungsoo Lee, Jeonghoon Park, Daeyoung Kim, Juyoung Lee, Edward Choi, and Jaegul Choo. Revisiting the importance of amplifying bias for debiasing. AAAI 2023, pp. 14974–14981.
> [8] Zhilu Zhang and Mert Sabuncu. Generalized cross entropy loss for training deep neural networks with noisy labels. NeurIPS, 31, 2018.

---

### Author Response · Authors · 2025-11-21

We sincerely thank all the reviewers for their constructive feedback and for recognizing the strengths of our work, including its novelty in designing a multi-stage data filtering strategy to generate reliable pseudo-bias labels and in using the confusion matrix to mitigate bias, the conceptual improvements from coarse- to fine-grained distribution matching, the annotation-free design, and the robustness to multi-shortcut scenarios. These strengths reinforce the value and relevance of our approach.

Regarding the concerns about MST, although we do not provide new theoretical results, our method is grounded in research findings that are well acknowledged and widely accepted in the fields of debiasing and spurious-correlation mitigation. These include:
- ERM’s tendency to overfit spurious correlations;
- Bias-aligned samples generally exhibiting higher confidence;
- Improvements in bias prediction after removing bias-conflicting samples;
- The discovery of bias cues through violations of the Equal Opportunity Fairness (EOF) criterion.

Together, these established foundations offer strong support for the practicality and faithfulness of our method.

Some of Reviewer 3’s (FipK) concerns—such as the interpretability of the bias cues encoded in
$s$ and the use of ERM’s overfitting behavior to reveal them—appear to stem from limited familiarity with this research line. Several important experimental analyses also seem to have been overlooked, including the study on $\gamma$, the effectiveness of our method in complex multi-shortcut scenarios, the quantitative comparison between MST and XRM, and the impact of MST’s prediction quality on the subsequent FG-CCDB stage. Considering these points, the comments may not fully reflect the demonstrated contributions and effectiveness of our method.
We hope the final decision will take these points into consideration.

For ease of reading, we use the following abbreviations: for example, **R1W1** denotes the first weakness raised by Reviewer 1, and **R1Q1** denotes the first question raised by Reviewer 1. The main manuscript and the supplementary material have been revised accordingly.

---

### Meta-Review · Area_Chair_kai5 · 2026-01-03

**Summary:**

This paper introduces a novel debiased learning approach from the observation that existing class-conditional distribution balancing (CCBB) methods rely on simplistic single-Gaussian approximations. The paper proposes multi-stage data-selective reTraining (MST) and introduces fine-grained CCBB.

**Reviewer Concerns:**

The reviewers noted the method's novelty and empirical performance, but also raised the following concerns.

- Theoretical grounding (2Y79): Reviewer asks how iterative bias amplification improves minority-mode recall.
- Definitions (FipK): The reviewer mentioned that the definition of 'mode' is a rather black-box concept with unclear mapping to latent bias factors
- Comparisons (exPH, FipK): Lack of direct quantitative measures, lack of comparison to distinguish the novelty of MST
- Hyperparameter dependency (2Y79, FipK): weak justification on specific hyperparameters (selecting top-50% high-confidence samples)

The authors provided a rebuttal with the following arguments.
- Theoretical grounding: The authors mention that recent research findings support the validity of MST. The authors also explained how the idea is connected to Equal Opportunity Fairness (EOF)
- Definitions: authors clarified how the definition of 'mode' (s,y) had been made
- Comparisons: the authors added comparisons with labeling-free methods (DDB, DaC, GERNE) and show compelling performance. The authors also show that MST can capture all minority groups from UrbanCars dataset.
- Hyperparameters: the authors provided ablation studies on the top-50% selections
- In addition, the authors provided experiments with split modes using Waterbirds dataset using WGA (worst group accuracy).

**Reviewer Scores:**

The reviewers gave the following initial scores.

- 2Y79: Marginally above acceptance threshold
- exPH: Marginally above acceptance threshold
- FipK: Marginally below acceptance threshold

The reviewers' concerns and authors' rebuttal are summarized in Reviewer Concerns. AC confirms that most of the concerns are appropriately addressed. Given that reviewers agreed on the technical contribution and effectiveness of the proposed MST, and that the concerns raised by reviewers are adequately addressed in the rebuttal phase, AC recommends acceptance of the paper.

---

### Decision · Program_Chairs · 2026-01-26

Accept (Poster)